# Position: Bridge Human Interpretation and Machine Representation with Explicit Specification for Qualitative Data Analysis In LLM Era

**Xinyu Pi** [* 1]   **Qisen Yang** [* 1]   **Chuong Nguyen** [* 1]   **Hua Shen** [2 3]

## Abstract

Large language models are increasingly used to analyze human feedback, agent traces, evaluation rationales, and other unstructured records central to modern ML systems. Yet the field lacks a shared way to specify what kind of qualitative reasoning an LLM-based pipeline is intended to perform, what representation it should produce, and how such outputs should be evaluated. This position paper proposes an explicit specification perspective: separating meaning-making from modeling, and making both visible as part of the analytic. We introduce a 4×4 landscape that crosses levels of meaning-making with levels of modeling, and use it to situate and compare qualitative outputs across both human-led studies and LLM-assisted workflows. A structured analysis of 300 papers suggests that many current LLM pipelines emphasize surface organization and static representations, with fewer systems making explicit commitments to richer causal or dynamical models. We conclude with a research agenda for LLM-assisted qualitative analysis focused on explicit level selection, evidence-linked outputs, and governance mechanisms aligned with the strength of semantic and representational claims. We also release our annotation results[1].

## 1. Introduction

Qualitative data analysis, or qualitative research (QR), is a family of methods for constructing defensible meaning from text, discourse, and other rich records of human and system

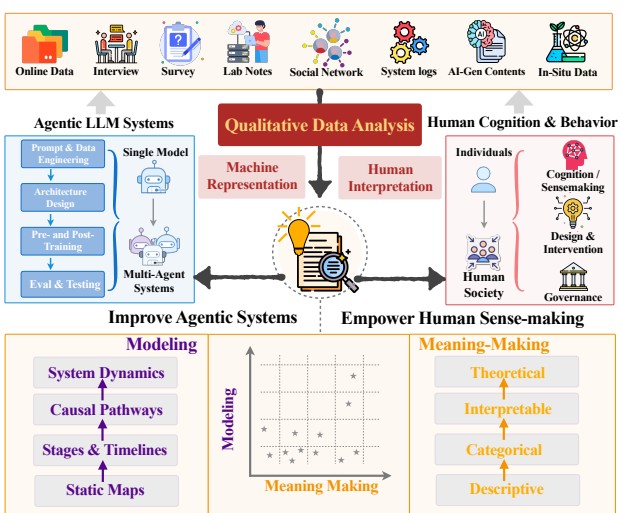

*Figure 1.* The Synergy of Qualitative Research and LLM developements in broader scientific context.

activity ([Flick, 2013](#)). Rather than reducing evidence to predefined scalar measurements, QR is designed for settings where the relevant structure is latent, context-dependent, and only partially observable. Historically rooted in the social sciences, QR has become increasingly relevant to contemporary AI research, where large-scale unstructured corpora (human feedback, system traces and conversations) are now central objects of study ([Ouyang et al., 2022](#); [Cemri et al., 2025a](#); [Tamkin et al., 2024](#)). Across AI and adjacent domains, researchers face a common challenge: making sense of messy, high-volume traces where neither the variables nor the mechanisms are known in advance. For example, to train better LLMs, modern RLHF algorithms rely on human preference data for alignment([Ouyang et al., 2022](#)), but it's hard to scale up without clear rubrics. When constructing agentic systems, people also need to go through inference records to find out where the power of multi-agent systems is and why they fail ([Cemri et al., 2025a](#); [Pan et al., 2025](#)). Human-centered AI interprets user interviews to systematically understand how LLMs are used in users' daily life ([Tamkin et al., 2024](#); [Huang et al., 2025](#)). In these settings, meaning must be constructed, justified, and iteratively refined. QR provides a principled framework for this end.

[1]University of California, San Diego, La Jolla, CA, USA [2]New York University Shanghai, Shanghai, China [3]New York University, New York, NY, USA. Correspondence to: Xinyu Pi <xpi@ucsd.edu>, Qisen Yang <qsyang@ucsd.edu>, Chuong Nguyen <chn021@ucsd.edu>, Hua Shen <huashen@nyu.edu>.

*Proceedings of the 43rd International Conference on Machine Learning*, Seoul, South Korea. PMLR 306, 2026. Copyright 2026 by the author(s).

[1]http://chuongnguyen26.github.io/QA-PositionPaper

However, conducting QR remains difficult in practice. The central bottleneck is not access to data, but the labor of constructing and defending coherent meaning from it. Recurring challenges include: (i) aligning and iteratively revising a codebook across hundreds of annotation sessions while maintaining audit trails; (ii) extracting and reconciling latent mechanisms from heterogeneous artifacts (e.g. policies, reports, and logs) that operate at different units of analysis; and (iii) diagnosing long, branching system traces where failures are distributed across roles and time (Saldaña, 2021; Miles et al., 2014; Cemri et al., 2025a). These challenges closely mirror long last specification game (Grädel et al., 2003), showing supervised ambiguity and the difficulty of attributing outcomes to causes, but are rarely addressed with explicit methodological tools.

To address these barriers, researchers have explored various computational supports for QR (Ferreira Barros et al., 2024; Parfenova et al., 2025). Recently, LLMs offer a promising lever by combining scalable extraction with increasingly capable interpretation and reasoning (OpenAI, 2023), and have been applied to workflows such as qualitative coding and thematic analysis (Dai et al., 2023; Lam et al., 2024; Qiao et al., 2025; Pi et al., 2025). Despite this momentum, existing LLM-driven QR efforts exhibit two structural limitations: *(i)* Most systems treat qualitative analysis as *code induction plus aggregation*: they generate labels and themes, but stop short of inducing the structured representations that many QR outcomes ultimately reside on (e.g., phase models of causal pathways). *(ii)* QR is a family of methods with different analytic commitments and output forms—including interpretative phenomenological analysis (IPA), grounded theory, thematic analysis, discourse analysis, narrative analysis, and related traditions (Smith et al., 2009; Glaser & Strauss, 1967; Charmaz, 2014; Braun & Clarke, 2006a; Miles et al., 2014). Though clearly distincted, this fragmentation has encouraged siloed automation pipelines that obscure shared analytic primitives.

**We argue that current LLM-supported qualitative research is constrained not by model capability, but by an impoverished specification of analytic commitments, and that progress requires an explicit theoretical framework.** To make those commitments explicit in a way that can guide system design, we follow principles of modern system design (Farley, 2021): rather than having a bespoke pipeline for each species, we identify a small set of interoperable building blocks and a *specification language / design protocol* that makes a QR standard explicit—what kind of meaning-making it claims to commit, and what kind of model it claims to produce. Under this protocol, qualitative automation can be organized as a system of **high-cohesion, low-coupling** modules that **compose flexibly** to support diverse QR species.

To identify the recurring dimensions that organize qualitative research outcomes, we conducted an extensive analysis of hunndreds of prior QR papers, consulted domain experts, and converged on two condensed dimensions after 5 months' iterations (App. B). The first axis captures **level of meaning-making**, from surface description to theory-mediated inference. The second captures **level of modeling**, from static themes and relational maps to explicit mechanism and system-dynamics models (Sterman, 2000; Meadows, 2008). This yields our primary model of the design space: a **4×4 landscape** defined by crossing the 4 levels of meaning-making with the 4 levels of modeling.

A broad inspection of recent QR systems (including a structured review of **300** instances, see Fig. 2 for the whole overview) reveals a clear empirical point: most existing systems remain confined to regions that are insufficient for the kinds of *meaning-making* and *modeling* claims they implicitly promise – even when the motivating task would naturally call for deeper interpretation or richer structural explanations (Barros et al., 2024; Parfenova et al., 2024; Schroeder et al., 2025). This gap is not merely a matter of degree. Progressing from surface themes to interpretive claims requires explicit warrants for latent inferences, while moving from relational sketches to richer models demands concrete modeling commitments that specify how phenomena evolve, not just which concepts are linked.

We propose this landscape not as a taxonomy to be filled in, but as a diagnostic instrument that exposes where the community has systematically under-committed. In Sec. 5, we propose call for actions for both NLP/LLM researchers and qualitative researchers and provide suggestions on how QR analytic commitments should be specified and governed, which representational moves are needed to move beyond modeling-shallow outputs, and what kinds of artifacts and infrastructures are required to boost LLM-assisted QR research. The question is no longer whether LLMs can assist qualitative research, but whether we are willing to explicitly and rigorously specify what kinds of meaning and models we are asking them to produce.

## 2. From Unstructured Data to Structured Insights and Models

A central reason qualitative research (QR) is powerful yet difficult is that many target phenomena are *not directly observable as variables*. In organizational and sociotechnical settings, we often care about norms, coordination regimes, failure patterns, interaction dynamics, adaptation, and multiparty sensemaking that are system-level and latent. What we typically observe are *partial unstructured datum*: interviews, documents, fieldnotes, meeting minutes, chat transcripts, incident reports, and (in AI settings) interaction logs and tool traces (Cemri et al., 2025b). These traces are perspectival,

incomplete, and context-bound. Like the "blind men and the elephant," each trace touches only part of an underlying mechanism, and apparent inconsistencies can become coherent only after within-frame interpretation and integration.

This challenge is especially salient for modern LLM deployments. LLM systems become multi-agent, tool-mediated, and long-horizon, the object of analysis is less a single input–output mapping and more an evolving interaction process (Wu et al., 2023; Qian et al., 2024; Hong et al., 2023). Consequently, diagnosis often takes the form of qualitative analysis over long traces (who believed what, when coordination broke, why verification failed). The outputs are frequently QR artifacts such as failure typologies, phase narratives, and feedback explanations (Cemri et al., 2025a). Methodologically, "conducting QR" is therefore not merely *summarizing text*. It is a disciplined attempt to reconstruct latent structure from fragmentary evidence.

## 2.1. A quantitative–qualitative analogy

To make this reconstruction problem legible to NLP/ML audiences, we contrast QR with a classical quantitative modeling pipeline. In quantitative modeling, the analyst often begins with a specified observation space (variables/features) and a model family, and success is anchored by an explicit fit criterion (likelihood, loss, $R^2$) (Box, 1976). In QR, the observation space is often *textual* and the latent variables are not given. Analysts must *induce* semantic units and justify the inferences that link excerpts to claims about the underlying phenomenon (Miles et al., 2014; Saldaña, 2021; Lincoln & Guba, 1985). This is why QR emphasizes traceability (how claims connect to excerpts), credibility/trustworthiness, and iterative refinement of concepts (Krippendorff, 2018; Graneheim & Lundman, 2004).

The analogy is imperfect, but it highlights a computational point. QR requires making two kinds of commitments explicit: what meanings are inferred *from and across traces*, and what global structure is asserted *over those meanings*. In case-based theory building, for example, researchers emphasize disciplined within-case interpretation and careful cross-case synthesis, rather than optimization of a scalar objective (George & Bennett, 2005).

## 2.2. Two coupled inference problems

The qualitative research community has already built a relatively comprehensive taxonomy of qualitative analysis methods, yet a gap remains between human inquiry and machine automation—especially when the underlying phenomenon is only indirectly visible through traces. We therefore argue that qualitative analysis must solve two coupled inference problems: one about meaning (both within and across traces) and one about structure (how inferred meanings are organized into system-level claims).

| Feature | Quantitative data analysis | Qualitative data analysis |
|---|---|---|
| Input data | Numeric/tabular observations | Unstrucutred data (interviews, docs, logs) |
| Primary inference | Parameter estimation, hypothesis testing | Meaning inference + structural reconstruction and integration |
| Model products | Regression, classifiers, ODEs | Themes, stage models, causal graphs, feedback systems |
| Eval target | Fittness / predictive loss | Traceability and consistency to corpora |

*Table 1.* A motivating analogy between quantitative and qualitative analysis. Both involve modeling under uncertainty, but differ in input modality, inferential focus, representational products, and evaluation criteria.

**(1) Meaning inference within and across traces.** Given an excerpt—and often by triangulating multiple excerpts across people, times, or artifacts — what can we *legitimately infer* beyond surface form? This includes deciding whether we are describing what is said, organizing it into categories, inferring implicit commitments entailed by the situation, or applying an external conceptual lens (Berelson, 1952; Hsieh & Shannon, 2005; Gioia et al., 2013; Krippendorff, 2018). This meaning-making can happen cross-trace (e.g., resolving apparent contradictions, refining category boundaries, or semantically aggregating shared patterns).

**(2) Across-instance structural reconstruction.** Given many such inferences, what *kind of system-level structure* are we claiming? A study may synthesize traces into a thematic map, a phase model, a mechanism diagram, or a dynamical feedback explanation, and these end products impose different validity demands (Miles et al., 2014; George & Bennett, 2005). Adjacent traditions make the same point with different formalisms, including process tracing (mechanism evidence), mechanism-based explanation (generative processes), and configurational approaches such as QCA (combinations of conditions) (Bennett & Checkel, 2015; Hedström & Swedberg, 1998; Ragin, 1987).

## 2.3. Why we factorize into meaning-making and modeling

These two inference problems motivate a coordinate system rather than a single catch-all label of "analysis." *(i)* We need a **level of meaning-making** axis to state how strong the semantic leap is at the level of evidence interpretation: paraphrase vs. categorization vs. within-frame interpretation vs. theory-mediated reframing within or across traces. Without this axis, surface-level reorganization can be conflated with analysis involving substantive interpretation or theoretical commitment. *(ii)* We need a **level of modeling** axis to state what form of system claim is made at the *phenomenon level*:

static relational configuration, staged process, causal dependency structure, or evolving feedback system. Without this axis, "building theory" collapses into one bucket, even though a taxonomy, a phase model, a causal graph, and a dynamical system model are qualitatively different products with different criteria for warrant and evaluation (Zeigler et al., 2000; Law, 2015; Sterman, 2000).

Taken together, the two axes reduce category errors (e.g., mistaking thematic coding for a mechanism explanation), align evaluation with the claims being made, and sharpen where current LLM capabilities lie. We therefore introduce the four levels of meaning-making in Sec. 3.1 and the four levels of modeling in Sec. 3.2, and use their cross-product as the organizing map for LLM-driven qualitative analysis.

## 3. Landscape Model: Describing Qualitative Analysis Commitments

Qualitative analysis reconstructs latent phenomena from *partial textual traces* (interviews, documents, logs, incident reports, transcripts) by making two kinds of commitments: (i) what is inferred from any given excerpt, and (ii) what global structure is asserted across excerpts (Miles et al., 2014; Saldaña, 2021; Lincoln & Guba, 1985). We factorize these commitments into two analytically separable axes. The first axis is a **level of meaning-making** (how far the analysis moves beyond surface-faithful restatement toward interpretation and theory-mediated reframing), drawing on classic manifest/latent and emic/etic distinctions (Berelson, 1952; Krippendorff, 2018; Graneheim & Lundman, 2004; Pike, 1954). The second axis is a **level of modeling** (what explicit representational structure is produced: static maps, stage models, causal mechanisms, or dynamical feedback systems), drawing on system science and simulation traditions (Box, 1976; Zeigler et al., 2000; Law, 2015).

Importantly, two axes are analytically separable: the same interpretive claim can be expressed as a thematic map, a phase narrative, a causal diagram, or a feedback model. Their cross-product yields a **4×4 landscape** that makes QR products comparable across methodological "species" and clarifies why evaluation (and LLM-based automation) must be aligned to the kind of claim being made.

### 3.1. Dimension 1: Level of Meaning-Making

We use *meaning-making* to describe **what an analysis adds beyond the literal statements**. At low levels, analytic outcomes stay trace-faithful. At higher levels, outcomes introduce stronger *semantic commitments*: they infer latent intent and causations, and may further re-interpret the case through an external theoretical lens. We distinguish 4 levels of meaning-making by whether the output commits to (i) **implicit inference** beyond what is explicitly stated, and

(ii) **external conceptual resources** beyond the case's own frame. We only briefly discuss our definition here and give a more rigorous articulation in App. D due to page limit.

**M1: Descriptive (manifest, surface-faithful). M1** outputs restate, summarize, or extract *explicit* content with minimal abstraction and minimal inference. This aligns with "manifest" content analysis: what is directly observable in the record and checkable against the source (Berelson, 1952; Holsti, 1969; Krippendorff, 2018). In computational terms, **M1** corresponds to grounded summarization and information extraction when the output remains trace-faithful.

**M2: Categorical (themes, topics, patterned organization). M2** outputs introduce *organizational abstraction*: they group observations into recurring categories/themes/types (a reusable label system) while remaining primarily *frame-faithful*—they organize what appears in the corpus more than asserting deeper latent "why" claims (Braun & Clarke, 2006a; Miles et al., 2014; Saldaña, 2021). Computationally, **M2** aligns with thematic summarization, taxonomy induction, and clustering/topic-organization when the result is an organization of surface content rather than an inference about implicit commitments.

**M3: Interpretive (implicit meaning entailed by the frame). M3** outputs articulate *implicit or latent meaning* that is not explicitly stated but is plausibly *entailed by the situation's semantic frame*—e.g., underlying goals, obligations, causal mechanisms, hidden patterns, pragmatic intent, or unstated constraints that make the episode coherent (Graneheim et al., 2017). This level is grounded in (i) **frame-based inference** (Fillmore, 1982; Baker et al., 1998), (ii) **pragmatics and implied commitments** (Grice, 1975; Searle, 1969; Stalnaker, 2002), and (iii) interpretivist accounts of sensemaking and thick description (Goffman, 1974; Weick, 1995; Geertz, 1973). For NLP readers, **M3** is closest to entailment-style inference in spirit (Dagan et al., 2006; Bowman et al., 2015): the claim is *defeasible* but should be warranted by cues and shared background assumptions appropriate to the case.

**M4: Theoretical (external reframing; etic constructs). M4** outputs re-situate the material inside an *external theoretical or conceptual system* that is not licensed by the frame alone (etic vocabulary, theoretical coding, sensitizing concepts). The defining criterion is that the imported framework functions as an *analytic engine* shaping what counts as evidence and how claims are constructed (Hsieh & Shannon, 2005; Gioia et al., 2013; Saldaña, 2021). This includes theory-guided seeing and abductive/theory-mediated reinterpretation (Blumer, 1954; Tavory & Timmermans, 2014), as well as productive interpretation in hermeneutic traditions (Gadamer, 1989; Ricoeur, 1976). Because **M4** introduces stronger conceptual scope, it raises higher demands for warrant and explicit limits.

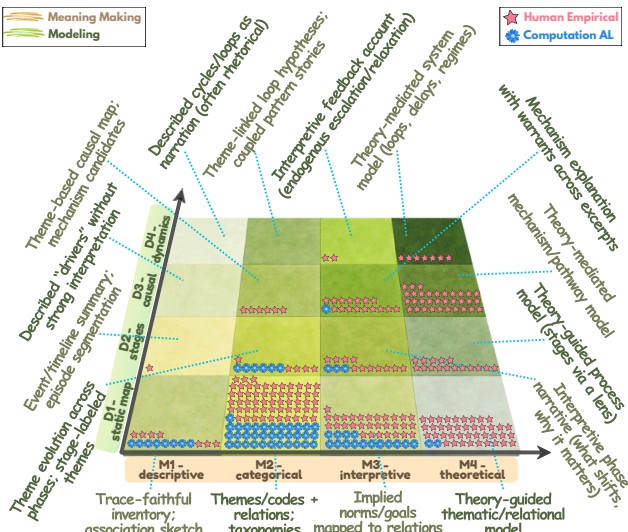

*Figure 2.* **QR Landscape**: Our $4 \times 4$ landscape model and annotated paper distribution.

## 3.2. Dimension 2: Level of Modeling

Here, *modeling* refers to the **explicit representational structure** produced to describe a phenomenon: what the units are, what relations hold among them, and what kinds of reasoning the representation supports (models as representations, not necessarily predictive estimators) (Box, 1976; Zeigler et al., 2000; Law, 2015). We distinguish 4 levels by whether the model commits to (i) **time**, (ii) **causality/mechanism**, and (iii) **system dynamics** (loopy iterative state updates and feedback). An expanded discussion can be found in App. E.

**D1: Static taxonomy and relational configuration models.** **D1** models are **static**: they represent themes, constructs, topologies and relationships without committing to temporal evolution or causal production. Typical outputs include thematic maps/networks (Braun & Clarke, 2006a; Attride-Stirling, 2001), cross-sectional relational/network representations (Wasserman & Faust, 1994; Borgatti et al., 2009), and concept maps used to organize relations without time/causality semantics (Trochim, 1989; Novak & Cañas, 2008). Signature: "what relates to what" in a snapshot.

**D2: Stage/phase/timeline models (time without causality).** **D2** models make **time** first-class: the phenomenon is organized as stages, episodes, or a timeline, but edges primarily mean "next/then" rather than "produces/changes" (Van de Ven & Poole, 1995; Langley, 1999; Pettigrew, 1990). Related sequence-analytic descriptions belong here when the primary commitment is ordering and segmentation rather than mechanism (Abbott, 1995; 2001).

**D3: Causal dependency and mechanism models (directed influence).** **D3** models explicitly represent **directed influence**: nodes are constructs/states/actors and edges encode causal/mechanism-like dependencies (en-

able/constraint, produce/change, mediate) (Pearl, 2009; Spirtes et al., 2000). This includes qualitative causal maps and fuzzy cognitive maps when used as influence structures (Kosko, 1986). Signature: "why/how" as directed production, typically still read as a (possibly qualitative) dependency graph rather than an iterated dynamical system.

**D4: Dynamical systems / feedback / complex systems models (state + update + iteration).** **D4** models treat the phenomenon as an **evolving system** whose behavior depends on **iterative state change** (state at $t$ shapes state at $t+1$), often via feedback, delays, and nonlinear responses (Forrester, 1961; Sterman, 2000; Meadows, 2008). This includes qualitative system-dynamics reasoning (reinforcing/balancing loopy machinery), coupled state-transition accounts with endogenous evolution, and simulation models (agent-based, discrete-event) when used to explain emergent trajectories (Bonabeau, 2002; Macy & Willer, 2002; Epstein, 2006; Gilbert & Troitzsch, 2005). Signature: the core explanatory claim depends on iteration and feedback, not just a static pathway diagram.

### 3.3. The integrated 4×4 conceptual landscape

Figure 2 integrates our two axes—**meaning-making** (rows) and **modeling** (columns)—into a $4 \times 4$ landscape that functions as a specification surface for qualitative outputs. Each cell corresponds to a distinct *level of semantic commitment* and a distinct *level of representation*: moving right increases semantic commitment (from surface-faithful description to external-lens theorizing), while moving upward increases representational commitment (from static relational structure to time, causal dependence, and system dynamics). With this decomposition of qualitative research commitments, we can situate different qualitative research traditions (e.g., content analysis, thematic analysis, grounded theory, IPA) within the landscape. We present detailed methodological mappings later in Appendix C due to page limit.

**Why the landscape matters for automation and evaluation.** The landscape prevents category errors (e.g., mistaking strong thematic coding for a mechanism explanation) and makes explicit that "good" is cell-dependent: **M1** is judged primarily by trace fidelity; **M2** by category coherence and coverage; **M3** by inference warrant and stability under alternative readings; **M4** by appropriateness and disciplined application of the external lens (Lincoln & Guba, 1985; Graneheim & Lundman, 2004). On the modeling axis, **D1** is judged by structural organization; **D2** by temporal fidelity and boundary clarity; **D3** by mechanism evidence and directional warrant; **D4** by whether feedback/iteration is genuinely supported (loops "close" with reciprocal evidence) (Sterman, 2000; Pearl, 2009).

### 3.4. Worked Example: Multi-Agent Failure Traces

Consider a corpus of multi-agent LLM system traces in which a task fails after decomposition, retrieval, synthesis, and verification. At D1, an analysis might produce a static taxonomy of failure types, such as planning drift, verification omission, and tool misuse. At D2, the same traces could be organized into a stage model: task decomposition $\rightarrow$ retrieval $\rightarrow$ synthesis $\rightarrow$ verification, showing where failures tend to arise. At D3, the analysis would propose a mechanism, such as: ambiguous decomposition creates inconsistent subgoals, which weakens retrieval checks and produces unsupported synthesis. At D4, the analysis would model task framing, retrieval quality, verification pressure, confidence, and coordination load as evolving state variables whose feedback structure explains repeated failure patterns.

Holding the modeling level fixed at D3, the meaning-making level changes the semantic commitment. M1 would state that retrieval was skipped before the final answer. M2 would categorize recurring causal patterns across traces. M3 would infer that the system implicitly prioritized answer completion over verification. M4 would reinterpret the mechanism through an external lens such as distributed cognition, coordination failure, or organizational sensemaking.

### 3.5. Operationalizability Experiment

As a limited validation of the operationalizability of our theoretical framework, we randomly sampled 35 papers from the 300-paper audit corpus and conducted an independent annotation pass using GPT-5.2. The model received only the level definitions and boundary rules used in our human annotation protocol, without access to human labels. We then compared model assignments against the human consensus labels. This experiment is intended to test whether the rubric is explicit and decision-consistent enough to be applied by an independent annotator; it is not a benchmark of LLM qualitative-analysis capability. Exact-match agreement was 85% for meaning-making and 94% for modeling (Appendix H). Cohen's $\kappa$ was 0.744 for meaning-making and 0.906 for modeling, corresponding to substantial and almost-perfect agreement, respectively. These results suggest that the rubric is operationally legible and decision-consistent in this limited setting. They should not be interpreted as evidence that GPT-5.2 can robustly conduct qualitative analysis at scale.

## 4. The Empirical Gap: What's Systematically Under-Committed?

To provide diagnostic evidence for our position, we conducted an audit of 300 papers. We do not treat this corpus as an unbiased census of qualitative research or computational QR automation. Instead, we use it to characterize visible patterns in how qualitative outputs are currently specified and represented. This empirical check prevents the landscape from becoming a purely conceptual taxonomy and provides an evidence-backed view of where existing work concentrates. We curated two complementary paper sets that reflect the two communities our framework connects – the **empirical QR papers** and the **computational QR papers**:

### 4.1. Paper Collection Processes

**Empirical QR papers** refer to classical qualitative research in which human researchers manually code, conduct narrative synthesis, and build models from empirical records. We curated papers using a **PRISMA-style staged screening procedure** (identification, deduplication/screening, eligibility assessment, and inclusion). We compile 4,493 domain keywords across 51 domains and paire them with 10 QR methodology terms (e.g., grounded theory, thematic analysis, narrative analysis), yielding 44,930 query combinations of the form `<domain keyword> AND <QR method>`. Using Google Scholar, we retrieve the first **3 pages** of results per query and obtained 664,244 records (identification stage). We then remove duplicates and retaine only entries with downloadable full texts, resulting in 138,683 downloaded papers (screening stage). Next, we perform an eligibility triage using **Qwen3-32B** to retain papers that report empirical qualitative studies with no heavy reliance on AI automation (lightweight HCI tools such as NVivo were allowed); we exclud purely computational papers, theoretical papers, and literature reviews, yielding 32,321 eligible candidates. Finally, we draw a **simple random sample** of 231 papers from this pool for manual annotation (inclusion stage). Additional details—including the full keyword list, exact prompts/filters, and exclusion breakdown—are provided in the Appendix.

**Computational papers** employ computational techniques to automate some major part of QR processes. In our analysis, we focus exclusively on the meaning-making and modeling responsibilities assumed by LLM-based components, abstracting away from human analytical steps outside the automated pipeline. Unlike empirical QR, few NLP or AI papers explicitly self-identify as automating qualitative research, which makes purely keyword-driven systematic retrieval insufficient. Accordingly, our curation process for computational papers adopts an **exploratory, human-in-the-loop literature expansion strategy based on snowball sampling** (Biernacki & Waldorf, 1981). We begin with a set of seed queries (e.g., `LLM Thematic Analysis`, `Automatic Schema Induction`, `Event Causal Graph Construction`), `Automatic Taxonomy Construction` and iteratively expand the corpus through backward and forward citation tracing. Papers are manually assessed and retained only if they propose *novel*

*automation pipelines* for inducing patterns or structures from unstructured data, rather than merely applying off-the-shelf tools. Following this process, we identify and retain 69 eligible computational papers for annotation. We acknowledge that our corpora is conditioned on query terms and visibility, and the distributions might not be perfectly unbiased population estimates.

## 4.2. The Paper Annotation Process

We manually annotated all sampled papers according to the operational definitions of the meaning-making axis (**M1**–**M4**) and the modeling axis (**D1**–**D4**) introduced in Sec. 3. Three authors served as annotators. Each paper was independently annotated by at least two annotators on *both* axes, using a shared rubric that specifies level definitions, boundary tests, and decision rules (Appendix G). Annotators were instructed to judge the paper's *dominant analytic output* and *dominant substantive model of the target phenomenon* (rather than incidental examples, background discussion, or the computational pipeline itself), and to ground their decisions in concrete representational features (e.g., what nodes/edges correspond to; whether structure is source→sink pathway-like vs. feedback-coupled; whether meaning claims remain surface-faithful vs. introduce implicit commitments vs. import external theoretical constructs). Disagreements were resolved through discussion among the annotators, resulting in a single consensus label per paper for each axis. We report the final consensus annotations in Fig. 2, and release the full rubric, prompts, and boundary heuristics used for annotation in the Appendix G to support reproducibility.

## 4.3. Observations

Three consistent patterns can be observed from Fig. 2, which together reveals the gap between LLM-based and traditional QR practices.

**(O1) Sampled computational work is modeling-shallow.:** They concentrate heavily in **D1**–**D2**, with only sparse coverage of **D3**–**D4**. Many systems do progress along the *meaning-making* axis (from descriptive toward categorical and occasionally interpretive outputs), but this semantic enrichment rarely translates into stronger *representational commitments*. Even when outputs contain richer semantics, they are typically encoded as static artifacts (e.g., themes, relational maps) rather than mechanism-bearing models.

**(O2) Sampled human QR is meaning-deep but often modeling-light.**. Empirical QR papers are markedly right-shifted on *meaning-making*, with most studies occupying **M3**–**M4**. However, many of these papers still instantiate their claims in **D1**–**D3** rather than **D4**. In other words, human QR frequently makes strong interpretive or theory-mediated commitments without necessarily formalizing them into explicit feedback- or state-update representations.

**(O3) In the sampled empirical corpus, high-level modeling is usually paired with high-level meaning-making.**: In the empirical corpus, high-level modeling almost never appears with low-level meaning-making: the upper-left region (high modeling, low meaning) is essentially empty. When papers reach **D3** (causations) or **D4** (dynamics), they almost also make **M3**–**M4** commitments, suggesting that richer representation typically require (and are justified by) deeper interpretive grounding.

## 5. Call For Action

The observed gaps from Fig. 2 motivates an agenda for both NLP/LLM and qualitative researchers. Agenda operationalization details are in App. I.

## 5.1. Implications for NLP/LLM researchers

For NLP/LLM researchers building systems for qualitative automation, the landscape suggests three directions:

**(1) Design code relation induction algorithms** that go beyond semantic aggregation of codes by making explicit the transition from *coding individual data points* to *modeling relationships among codes*. In QR terms, this corresponds to moving from open coding (meaning-making over individual instances) to axial coding, where analysts examine how codes relate through causal, temporal, logical, or interactive relations. Most existing computational work effectively stops at **D1** because they treat codes as independent labels or clusters and does not attempt to induce structured relationships among them. Advancing to **D2**–**D4** therefore requires methods that explicitly reason over inter-code relations rather than merely condensing code semantics.

**(2) Enable adaptive level selection**, i.e., mechanisms that let the system decide which level of meaning-making and modeling is warranted by a given corpus, with calibrated abstention or fallback to lower-commitment outputs when evidence is insufficient or ambiguity is high.

**(3) Create evaluation methodologies for qualitative modeling** that operationalize "fitness-to-corpus" for **D2**/**D3**/**D4** structures (e.g., evidence coverage, contradiction rate, pattern matching and semantic consistency), since humans perform such assessment intuitively but systematic approximations remain unclear. Beyond alingment with human ground truth models, automatic goodness-of-fit metrics for qualitative models will be a key missing ingredient for scalable automation and rapid research iterations.

These directions should not be read as a uniform call for all systems to target D4. We view the levels as progressively stronger commitments with different feasibility profiles. Near-term systems can more reliably support M1–M2

and D1–D2 tasks such as trace-faithful extraction, coding, categorization, and temporal segmentation. Medium-term systems may support M3 and D3 outputs when they are evidence-linked and include contradiction checks, counterexamples, and human adjudication. D4 should be treated as an upper-bound representational target: appropriate only when the corpus genuinely supports feedback, recurrence, state updates, and temporal iteration.

Feasibility and evaluation should therefore be treated as level-dependent rather than a single automation target. Near-term LLM systems are likely to be most reliable for lower-commitment operations such as M1–M2 extraction, coding, categorization, D1 static organization, and D2 temporal segmentation. M3 and D3 outputs require stronger safeguards, including evidence-linked rationales, contradiction checks, and human adjudication of implicit claims and directed mechanisms. D4 should be treated as an upper-bound representational target rather than a default goal for current systems: it is appropriate only when the corpus genuinely supports feedback, recurrence, state updates, and iterative temporal change. Accordingly, evaluation should match the claimed modeling level: D1 outputs should be assessed for structural coherence and coverage; D2 outputs for temporal ordering and boundary fidelity; D3 outputs for evidence-linked mechanism support, directional warrant, and counterexample search; and D4 outputs for whether feedback loops and iterative-update claims are actually supported by the traces. For example, an induced D3 graph should not be judged by surface plausibility alone, but by whether each proposed edge has excerpt-level support, and checks against reversed or contradictory cases.

### 5.2. Implications for qualitative researchers

For qualitative researchers and domain experts, the landscape should be treated less as a "taxonomy" and more as a **specification language for analytic contracts**: it clarifies what kind of output is being requested, what counts as acceptable warrant, and which risks are tolerable. We highlight the following directions:

**(1) Specify QR analytic commitments** by explicitly specifying the intended meaning-making and modeling levels for each analytic step (e.g., D1 theme condensation versus D3 mechanism proposal), thereby clearly distinguishing descriptive organization from interpretive or theoretical claims and improving transparency, comparability, and accountability across studies and tools.

**(2) Design governance rules to mitigate risks of LLM-assisted QR.** Empirical studies and practitioners report consistently emphasize the risks of LLM-assisted qualitative research, including overconfident fabrication, erosion of contextual grounding, and misalignment with qualitative values and human judgment (Schroeder et al., 2025; Barros et al.,

2024; Parfenova et al., 2024; Übellacker, 2024; Jowsey et al., 2025). However, one plausible hypothesis is that these risks might not distribute uniformly across all kinds of qualitative work. Not all meaning-making and modeling operations are equally demanding or equally fragile. It is plausible that LLMs can perform lower-level tasks (e.g., M1–M2, D1–D2) with relatively high reliability, while error rates, epistemic drift, and value misalignment increase as semantic and representational commitments deepen. By explicitly delineating which regions of the landscape are appropriate for automation (often M1–M2, D1–D2) and which require sustained human adjudication and positionality-aware interpretation (often M3–M4, D3–D4), the question shifts from a binary "should we use LLMs" to a scoped governance decision about when and where automation might be appropriate. Our framework could conceptually scaffold systematic benchmarking to establish more rigorous governance rules for applying LLMs in QR.

**(3) Create shareable gold artifacts that make qualitative standards testable**—including curated codebooks *(the collection of corpora-induced meaning units)*, explicitly documented boundary and counterexamples, and trace-linked rationales that connect excerpts to analytic claims. Primary qualitative materials (e.g., interviews, transcripts) and their associated codebooks are often closed due to privacy, confidentiality, and ethical constraints, which limits direct reuse for evaluation. As a result, the lack of standardized benchmarks that pair raw corpora with aligned, authoritative codebooks forces computational works to rely on bespoke datasets and derived codebooks for evaluation (Pi et al., 2025; Zhong et al., 2025; Qiao et al., 2025). Without human ground truth interpretation, the robustness of LLM self-evaluation for meaning-making is questionable. Moreover, QR outcomes are typically narrative and discursive, making them poorly suited for evaluating model fitness or structural adequacy. Converting core narrative claims into crisper representational forms—such as relational graphs, mechanism maps can enable systematic comparison across different QR pipelines.

## 6. Alternative Views

Some qualitative traditions treat early formalization as a liability. Exploratory, inductive, and abductive work often relies on deliberate openness: analysts keep multiple provisional readings in play and let concepts evolve through immersion, reflexivity, and narrative synthesis. From this standpoint, explicitly specifying a meaning-making or modeling level can be seen as premature closure. It may freeze tentative interpretations, reify distinctions that are fluid in practice, and privilege what is easy to label or diagram over insights that remain contextual or narrative. Many traditions therefore treat analytic restraint and delayed formalization

as a mark of rigor, not a lack of clarity.

A second view is to organize the space around existing qualitative-method traditions rather than our two axes. Method-centered taxonomies—such as thematic analysis, grounded theory, IPA, discourse analysis, narrative analysis, framework analysis, and ethnography—are well established and remain essential for methodological training. However, method names do not always specify the strength of semantic inference or the representational form of the output. A grounded-theory study, for example, may produce categories, process models, or mechanism claims depending on how the analysis is conducted.

Another alternative is to use tradition-specific representational frameworks: process tracing and mechanism-based explanation for causal mechanisms, QCA for configurations of conditions, stage/process narratives for temporal development, and system dynamics for feedback. We view these not as competitors to be replaced, but as anchors that motivate regions of the landscape. Our two-axis framework is intended as a cross-cutting specification layer: it asks what kind of meaning-making and model are claimed and produced, regardless of which tradition the study uses.

We take this objection seriously, but we see a different primary risk in LLM-mediated workflows: unmarked escalation of epistemic authority. A pipeline can start with low-commitment operations (surface-faithful summaries, theme grouping) and drift into high-commitment claims (implicit intent, normative expectations, causal explanations) without any explicit decision that the evidence supports the stronger step. Because qualitative rigor depends on both how meaning is inferred from traces and how those inferences are assembled into a model of the phenomenon, escalation along either axis should be deliberate, documented, and contestable. Explicit specification is therefore an analytic contract: it records the intended strength of semantic and representational claims, the warrants required to justify them, and when stopping at a lower-commitment output is the responsible choice.

We introduce meaning-making and modeling only as an initial, minimal interface rather than a comprehensive theory of qualitative analysis. By separating semantic inference from representational structure, we allow analytic commitments to be explicitly chosen, adjusted, and governed, while preserving a core qualitative insight: the same interpretive claim can be legitimately rendered through multiple representational forms. We recognize that alternative decompositions of qualitative analytic commitments are possible, and we view this proposal as a foundation rather than a closure. Future work can elaborate, refine, or reconfigure this landscape on the basis we outline here.

Moreover, some researchers argue that LLMs should not participate in reflexive qualitative analysis at all (Jowsey et al., 2025). These critiques are well grounded within reflexive and interpretivist traditions: LLMs are trained on corpora with uneven social and historical coverage, and they can reproduce dominant interpretive frames, biases, and omissions embedded in those data. More fundamentally, LLMs lack lived experience, stable positionality, and the capacity for situated reflexive adjustment that human analysts bring to qualitative interpretation. We take these concerns as motivating, rather than undermining, the need for explicit specification. Because LLM capabilities are uneven, analytic commitments must be made visible: which forms of meaning-making are delegated to models, which require human reflexive judgment, and where abstention or handoff is the responsible choice. Explicit specification clarifies the boundaries of LLM involvement, supports auditable and contestable analytic decisions, and foregrounds gaps that demand caution or future methodological development. In this sense, specification functions as a governance mechanism for selective, constrained, and accountable use rather than an endorsement of unrestricted automation.

## 7. Conclusion

In this position paper, we show that progress in LLM-assisted qualitative research depends less on raw model capability than on explicit specification of analytic commitments. By disentangling meaning-making from modeling, the proposed 4×4 landscape provides a shared way for describing, comparing, and governing qualitative outputs across human and machine settings. Our empirical analysis shows that current systems systematically under-commit at the representational level, limiting their explanatory power. We argue that making semantic and modeling commitments explicit is a prerequisite for evaluation, responsible automation, and meaningful collaboration between qualitative researchers and LLM developers in the coming era.

## 8. Limitations

Our empirical audit is diagnostic rather than population-estimating. The empirical corpus is shaped by search visibility, query construction, full-text availability, and likely language bias; the computational corpus is shaped by seed selection and snowball-sampling visibility. The operationalizability experiment is also limited: it uses 35 sampled papers and evaluates rubric legibility, not general LLM ability to conduct qualitative research. Finally, D4 is intentionally defined strictly and should be interpreted as an upper-bound representational target, not as a near-term requirement for all LLM-assisted QR systems. Future work should expand to multilingual corpora, broader databases, book-length qualitative work, and domain-specific benchmarks.

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

# A. Related work

**Computational Qualitative Data Analysis Development**  Augmenting and automating qualitative data analysis has been investigated for a long time (Chen et al., 2016; Bryda & Costa, 2023). Existing works move from keyword matching to fully automation trial. CoAICoder (Gao et al., 2023) leverages AI to enhance human-to-human collaboration in qualitative data analysis. Cody (Rietz & Maedche, 2021) and PaTAT (Gebreegziabher et al., 2023) serve as a code recommender by learning from patterns in human-generated codes. These efforts first show chances in AI-assisted qualitative analysis. After that, researchers focus more on semantic ability and quality of computational methods.

Recently, Large Language Models have been used to facilitate qualitative research (Hou et al., 2024; Barros et al., 2025a). ThemeViz (Kang et al., 2025) and Barros et al. (Barros et al., 2025b) demonstrate that LLM can scaffold human efforts. At this stage, people primarily treat the LLM as a conversational partner. Building on this, several works aim to establish efficient human-AI collaborative frameworks for qualitative analysis (Rao et al., 2024; Sharma & Wallace, 2025; Wiebe et al., 2025; Xu et al., 2025). Similarly, MindCoder () introduces a framework designed to produce transparent analytical traces. From another branch, researchers also explored fully automated qualitative analysis workflow, performing initial trials on LLM-based automatic thematic analysis(Braun & Clarke, 2006b; Paoli, 2024; Khan et al., 2024). LLOOM (Lam et al., 2024), Thematic-LM (Qiao et al., 2025), HICode (Zhong et al., 2025) and Auto-TA (Yi et al.) provide fully-automated pipelines that convert an unannotated corpus to a comprehensive codebook. Consistent with the thematic-analysis paradigm, their goal is to summarize themes across the corpus, not to construct an explicit hierarchy of concepts or rigidly model logical relationships among codes (Braun & Clarke, 2021a). For methods with higher level modeling, they require explicitly hierarchical, relational, and iterative (e.g., grounded theory). LOGOS (Pi et al., 2025) tried to fully automated grounded-theory development, mirroring standard grounded-theory practice and support deeper and finer-grained interpretive analyses in qualitative research.

**Domain Specific Computational Qualitative Data Analysis**  Considerable attention has been devoted to qualitative analysis in recent years in computer science community (Li et al., 2020; 2023; 2022a; Jin et al., 2022; Wen et al., 2021; Du et al., 2022). A major line of work in the AI community represents events as standard knowledge graphs, where actions and named entities are modeled as nodes with predefined edge structures, and graph neural networks are used to perform reasoning over them. Adopting a comparable formalism, (Cheng et al., 2024) specifically addresses the domain of the electric vehicle battery supply chain. Software Engineering is also another area where qualitative analysis has been used (Treude, 2024; Montes et al., 2025). Moreover, the AI community also sees the potential of qualitative research in the trending area, like multi-agent systems (Cemri et al., 2025a; Pan et al., 2025), and vibe-coding (Huang et al., 2025). In the context of broader qualitative data analysis application, scholars across diverse disciplines are increasingly incorporating LLMs into their analytic pipelines. For example, policy analysts have utilized LLMs to distill themes from extensive text archives (Fang et al., 2025; Wang et al., 2025). Literature reviews, a cornerstone of qualitative inquiry, offer another fertile ground for LLM-driven analysis, specifically for detecting and structuring themes across large bodies of publication (Übellacker, 2024; Singh et al., 2025; Padmakumar et al., 2025).

**Evaluating Qualitative Data Analysis Results**  Existing frameworks for evaluating qualitative analysis emphasize a multi-dimensional approach designed to ensure both methodological rigor and theoretical depth (Nowell et al., 2017; Manuj & Pohlen, 2012; Charmaz & Thornberg, 2021; Ritchie et al., 2013; Gee, 2025; Smith et al., 2009). As synthesized in the literature, valid qualitative assessment relies on six primary pillars. (1) The analysis must demonstrate fidelity and empirical grounding, ensuring that the resulting theory is "true" to the data, captures the participants' voices, and possesses sufficient explanatory power. (2) The reasoning must be logical and sound, characterized by confirmability, analytic rigor, and interpretations that are strongly supported by evidence. (3) Consistency and stability are paramount; this requires dependability and traceability (an audit trail) to establish the transparency and reliability of the research process. (4) The reusability of the findings is evaluated through their transferability, allowing readers to judge applicability to other contexts via thick description. (5) The framework mandates sufficient reflexivity, requiring researchers to acknowledge their assumptions and maintain theoretical openness to modification. (6) The evaluation prioritizes the originality and contribution of the results, seeking novel conceptualizations that extend beyond local descriptions to offer generalizable theoretical understanding.

From the computational perspective, researchers also adpot different ways to evaluate qualitative analysis results. Traditional assessment typically depends on event/theme prediction tasks (e.g., edge type classification). People use traditional statistical scores, like F1, Recall, Precision, to measure how well the trained model can predict the event/theme (Li et al., 2020; Zhong

et al., 2025). When it comes to semantic level, some classic NLP measurements are used. BLUE, ROUGE, BERTScore are used to compare the overlap or similarisy between ground truth Model generated results (Parfenova et al., 2025; Yi et al., 2025; Qiao et al., 2025). However, given the rareness of human ground truth, research also explore more flexible way to judge the results, either using human judgement (Montes et al., 2025) or LLM-as-a-judge (Pi et al., 2025). But these evaluation methods are still in an early stage.

## B. How We Developed Our Theoretical Framework

### B.1. Overview and Design Stance

We developed the 4×4 landscape through an iterative theory-building process aimed at producing a framework that is (i) *conceptually clean* (non-overlapping commitments), (ii) *operational* (annotators can reliably apply it), and (iii) *interdisciplinarily grounded* (linked to established distinctions across qualitative methodology and modeling traditions). The process is best described as *grounded-theory-like*, but extended by the need to reconcile and anchor concepts across multiple foundational literatures (linguistics, pragmatics, interpretive social science, philosophy of science, systems science, and engineering design).

### B.2. Iterative Development Cycle

Across approximately five months, we conducted roughly ten rounds of iteration following a repeated cycle:

1. **Incremental paper collection and landscape exposure.** We expanded a seed set of representative works spanning (a) qualitative research methods and exemplars and (b) NLP/LLM-based qualitative automation and related modeling traditions. This expansion was iterative: newly encountered formalisms and analytic "moves" frequently revealed missing distinctions and edge cases, prompting further targeted collection.

2. **Pattern recognition and empirical grounding.** We repeatedly asked: *what kinds of meaning-making commitments the researchers attempt, and what kind of model form does the paper treat as its primary explanation?* We recorded recurring output commitments and representational primitives (e.g., surface-faithful restatement vs. categorical organization vs. within-frame inference vs. external theoretical reframing; and, for modeling, stage narratives vs. directed pathways vs. coupled-feedback systems).

3. **Pattern condensation into discriminative axes.** We progressively condensed observations into two orthogonal axes—**meaning-making level** and **modeling level**—with explicit boundary tests intended to reduce overlap and ambiguity. A key design goal was to classify *what the analysis commits to*, rather than *what the method is called*.

4. **Framework proposal, boundary testing, and revision.** We iteratively proposed definitions, then stress-tested them against challenging cases where boundaries are known to fail (e.g., L2 vs. L3 "themes vs. implied commitments"; L3 vs. L4 "within-frame inference vs. etic theoretical import"; modeling L3 vs. L4 "pathway-like directed dependence vs. mutually coupled feedback systems"). When definitions failed, we revised: (i) operational heuristics, (ii) decision rules, and (iii) failure-mode clarifications.

5. **Application-driven validation ("fitness" checks).** We repeatedly applied the evolving framework to diverse papers to check **coverage** (can it classify what exists?), **stability** (do similar papers land similarly?), and **operationality** (can others follow rules and reach comparable judgments?). These fitness checks served the same function as constant comparison in grounded theory: disagreements and hard cases were treated as signals that the framework required refinement rather than as annotator error.

### B.3. Interdisciplinary Theoretical Anchoring

Unlike single-paradigm framework construction, our development required explicit anchoring across disciplines to avoid purely rhetorical distinctions. For meaning-making levels, we aligned boundaries with established separations such as manifest/latent content, emic/etic distinctions, frame-based inference, and pragmatics (implicature, presupposition, and speech acts), as well as interpretivist sensemaking and thick description. For modeling levels, we grounded distinctions in the representational commitments of stage narratives, directed pathway models (e.g., DAG-like causal accounts, workflows/FSMs), and complex-systems interdependency models (feedback-centered coupling). This anchoring served two

| | Model L1 (static map) | Model L2 (stages/timeline) | Model L3 (causal pathways) | Model L4 (feedback dynamics) |
|---|---|---|---|---|
| **Meaning L1** (descriptive) | Trace-faithful inventory; association sketch | Event/timeline summary; episode segmentation | Described "drivers" without strong interpretation | Described cycles/loops as narration (often rhetorical) |
| **Meaning L2** (categorical) | Themes/codes + relations; taxonomies | Theme evolution across phases; stage-labeled themes | Theme-based causal map; mechanism candidates | Theme-linked loop hypotheses; coupled pattern stories |
| **Meaning L3** (interpretive) | Implied norms/goals mapped to relations | Interpretive phase narrative (what shifts, why it matters) | Mechanism explanation with warrants across excerpts | Interpretive feedback account (endogenous escalation/relaxation) |
| **Meaning L4** (theoretical) | Theory-guided thematic/relational model | Theory-guided process model (stages via a lens) | Theory-mediated mechanism/pathway model | Theory-mediated system model (loops, delays, regimes) |

*Table 2.* The 4×4 landscape as a conceptual coordinate system. Rows specify the semantic commitments made from traces; columns specify the representational commitments made about the phenomenon. Each cell implies different validity demands and evaluation targets.

purposes: (i) to justify why each level is qualitatively distinct rather than "just deeper," and (ii) to supply decision rules that are communicable to both QR and NLP audiences.

### B.4. Social Proof and Community Feedback

To ensure the framework is not only internally coherent but also intuitively usable by its intended communities, we repeatedly sought external feedback during development. We presented intermediate versions in a research seminar attended by 10+ graduate-level researchers (including cognitive science PhDs) and conducted informal expert consultations with more than 5 faculties and domain experts spanning philosophy, linguistics, and sociology. Across these interactions, feedback consistently emphasized that the framework is handy and intuitive as a shared vocabulary for describing (a) what an analysis output *commits to* and (b) what kind of model form it *privileges*. We used this feedback as a practical check on communicability: when experts found a boundary confusing or a criterion too implicit, we revised definitions and boundary tests accordingly.

## C. Mapping Qualitative Research Methods into the Landscape

The qualitative research community has developed a relatively comprehensive taxonomy of methods, addressing questions such as analytical targets, appropriate conditions of use, and conventions for data collection, processing and analyzing. Here, we map established qualitative research methods onto our analytical landscape. Table 2 show a conceptual coordinate metric for each (**Meaning making, Modeling**) pair. Fig. 3 generally shows how existing QR methods fit in our landscape. We emphasize that the placements below are **high-level and illustrative rather than exhaustive**. Method labels do not uniquely determine a single cell in the landscape; however, many research traditions exhibit typical or canonical regions of emphasis, arranged here generally from lower to higher degrees of interpretive and structural complexity:

Here, we provide a detailed mapping from traditional qualitative analysis methods to our landscape:

- **Qualitative content analysis / descriptive reporting** typically emphasizes **M1**–**M2** with **D1**, prioritizing faithful summaries, coding schemes, and relatively static thematic organization (Berelson, 1952; Hsieh & Shannon, 2005; Krippendorff, 2018).

- **Framework analysis** occupies a distinct position emphasizing **D2** with **M1**–**M2**, offering a structured, matrix-based approach to data management that facilitates systematic comparison across cases while maintaining a descriptive focus (Parkinson et al., 2016; Goldsmith, 2021).

- **Thematic analysis** commonly targets **M2** with **D1**, focusing on the identification and organization of themes; some variants extend toward **D2** when themes are explicitly sequenced or staged over time (Braun & Clarke, 2006a; Attride-Stirling, 2001). Highly reflexive thematic analysis can also go beyond to **M3** (Braun & Clarke, 2019; 2021b).

- **Interpretative phenomenological analysis (IPA)** typically aims at **M3**, and sometimes **M4** when explicit theoretical or philosophical lenses are central. Representationally, IPA often relies on **D1** or **D2** structures to prioritize the depth of

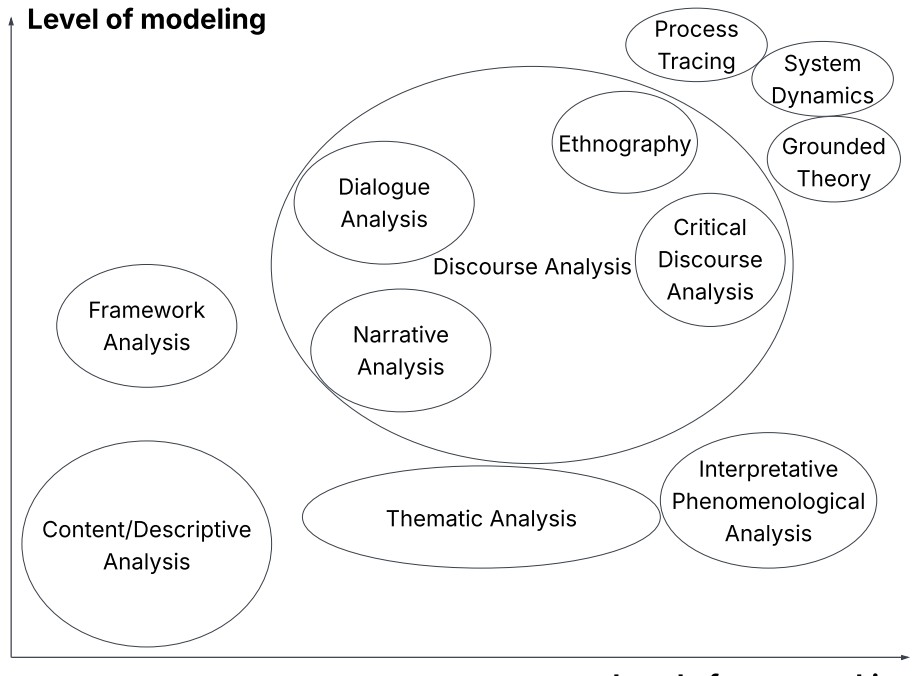

*Figure 3.* Mapping QR methods to our landscape

lived experience (Smith et al., 2009).

- **Discourse analysis and related traditions** (including **Narrative**, **Dialogue**, **Ethnography**, and **Critical Discourse Analysis**) form a central cluster in our landscape, frequently emphasizing **D2**–**D4** with **M2**–**M4**. These methods foreground the structures of communication, temporal bracketing, and cultural context (Brown & Yule, 1983; Langley, 1999). Within this cluster:
  - **Narrative analysis** and **Dialogue Analysis** foregrounds episodes and plots (Cortazzi, 1994; Carlson, 2012).
  - **Critical Discourse Analysis (CDA)** and **Ethnography** often push toward higher mean-making (**M3**) by integrating broader social power dynamics and cultural systems (Fairclough, 2023; Hammersley, 2006).

- **Process tracing and mechanism-centered explanation** align with **D3** and **M3**–**M4**, focusing on warranted within-case causal and mechanistic claims, often mediated by explicit theoretical expectations (Bennett & Checkel, 2015; Hedström & Swedberg, 1998).

- **Grounded theory and case-based theorizing** generally push toward **M3**–**M4** and often toward **D3**, emphasizing emergent mechanisms or explanatory logics, although the specific representational forms vary widely across traditions (Glaser & Strauss, 1967; Charmaz, 2014; George & Bennett, 2005).

- **System dynamics and simulation-based explanation** align most directly with **D4**, typically paired with **M3**–**M4**, as feedback-based and holistic explanations generally require substantial interpretive and theoretical commitments to model complex interdependencies (Forrester, 1961; Sterman, 2000; Meadows, 2008).

## D. Expanded Level of Meaning-Making

Qualitative research is often described as "interpretive," yet in practice different papers (and different LLM-based pipelines) construct meaning at markedly different depths and scopes. Some outputs remain close to what is explicitly said, some organize recurring topics, some infer implicit commitments and social/pragmatic implications, and some re-situate the same material inside external theoretical systems. Classic methods papers recognize related distinctions—for example, manifest versus latent content in content analysis (Berelson, 1952; Holsti, 1969; Krippendorff, 2018; Graneheim & Lundman, 2004; Hsieh & Shannon, 2005), or informant-centric versus researcher-centric "orders" of coding (Gioia et al., 2013; Saldaña,

2021). However, there is limited consensus on a single, domain-general way to characterize *what kind of meaning* an analysis produces, in a manner that is both (i) conceptually clean (non-overlapping levels) and (ii) operational (annotators can reliably distinguish levels).

We propose a four-level theory of meaning-making. Each level is defined by the *semantic commitments* introduced by the output, not by the qualitative method named in the paper. Concretely, we distinguish levels along two coupled dimensions: **semantic depth** (how much implicit meaning is made explicit) and **conceptual scope** (whether meaning is constructed within the case's own semantic frame, or by importing external conceptual resources). This yields four levels—Descriptive (L1), Categorical (L2), Interpretive (L3), and Theoretical (L4)—which we use as one axis of our 4×4 landscape.

### D.1. Level 1: Descriptive meaning-making (manifest, surface-faithful)

L1 outputs restate, summarize, or extract *explicit* content with minimal abstraction and minimal inference. This corresponds to "manifest" content in classic content analysis: what is directly observable in the text or record, in a way that can be checked against the source (Berelson, 1952; Holsti, 1969; Krippendorff, 2018). In qualitative content analysis terminology, L1 stays close to the surface form (e.g., who said what, what happened, what was mentioned), without interpreting intentions, unstated causes, or normative implications (Graneheim & Lundman, 2004; Kleinheksel et al., 2020).

For NLP/LLM readers, L1 aligns with extractive/abstractive summarization and information extraction when the output remains strictly grounded in the provided text (e.g., generating a faithful synopsis, listing mentioned entities, enumerating reported events).

### D.2. Level 2: Categorical meaning-making (themes, topics, and patterning)

L2 introduces *organizational abstraction*: it groups surface observations into recurring categories, themes, domains, or types. This is the core analytic product of many thematic and coding-oriented workflows (e.g., themes as patterned responses within a dataset) (Braun & Clarke, 2006a; Saldaña, 2021; Miles et al., 2014). Importantly, L2 abstraction is primarily *frame-faithful*: categories summarize what appears in the corpus rather than asserting a deeper latent "why" that must be true.

In content analysis terms, L2 often corresponds to building a coding scheme that clusters manifest observations into higher-level labels (Krippendorff, 2018; Hsieh & Shannon, 2005). For NLP/LLM readers, L2 aligns with clustering, topic modeling, weakly-supervised labeling, taxonomy induction, and thematic summarization—as long as the output remains an organization of surface content rather than an inference about implicit commitments.

### D.3. Level 3: Interpretive meaning-making (implicit meaning entailed by the frame)

L3 makes a qualitative shift: it articulates *implicit or latent meaning* that is not explicitly stated but is plausibly *entailed by the semantic frame* of the situation. This is closely aligned with "latent" content analysis in qualitative research (Graneheim & Lundman, 2004; Graneheim et al., 2017), while retaining a key constraint: the interpretation is warranted by the case's internal context and shared commonsense/genre knowledge, rather than by invoking a specialized external theory.

Three research traditions provide a principled foundation for this level:

**(i) Frame-based inference in linguistics and cognitive semantics.** In Frame Semantics, understanding an utterance requires evoking a structured frame (a schematic situation with participants and roles) that licenses inferences beyond what is overtly said (Fillmore, 1982). FrameNet operationalizes this idea for computational linguistics by cataloging frames and frame elements that support systematic within-frame inference (Baker et al., 1998).

**(ii) Pragmatic inference and implied commitments.** Classic pragmatics characterizes how speakers communicate more than they literally say, via implicature, presupposition, and speech-act force (Grice, 1975; Austin, 1962; Searle, 1969). Common-ground theory formalizes how discourse participants rely on shared background propositions that are not always explicit (Stalnaker, 2002). These mechanisms motivate L3 annotations such as inferring unstated goals, obligations, norms, or relational implications that make the discourse coherent.

**(iii) Interpretivist social science and sensemaking.** Interpretive traditions emphasize that meaning is constructed by connecting cues to contextual frames to form a coherent account (Weick, 1995; Goffman, 1974). "Thick description" explicitly treats interpretation as making implicit cultural/interactional context legible, without necessarily reducing it to

external theory (Geertz, 1973).

For NLP/LLM readers, L3 is the level most naturally connected to *natural language inference* and pragmatic reasoning: producing statements that are not verbatim in the text but follow from it under shared background assumptions (Dagan et al., 2006; Bowman et al., 2015). L3 outputs are defeasible (interpretations can be challenged), but they are constrained: they should be defensible as "what must (or very likely) is going on" *given* the situation described.

### D.4. Level 4: Theoretical meaning-making (external reframing and etic constructs)

L4 expands conceptual scope: it re-situates the case within an *external theoretical or conceptual system* that is not entailed by the semantic frame alone. The defining criterion is not whether a paper cites theory, but whether theory functions as an *analytic engine* shaping the meaning that is produced (Hsieh & Shannon, 2005; Gioia et al., 2013; Saldaña, 2021). In qualitative methodology terms, L4 aligns with moves such as theoretical coding and second-order theorizing, where the analyst introduces constructs that are not native to the participants' accounts (Gioia et al., 2013; Saldaña, 2021). This parallels the classic emic/etic distinction: L3 stays close to emic sense-making (within the case's frame), whereas L4 introduces etic categories (external analytic vocabulary) (Pike, 1954).

Two additional literatures motivate why L4 should be treated as a distinct level rather than "just deeper interpretation":

**Sensitizing concepts and theory-guided seeing.** Interpretive social science often uses sensitizing concepts to guide what counts as evidence and how observations are connected; such concepts are explicitly not reducible to surface description (Blumer, 1954). In modern qualitative theorizing, abductive analysis makes the theory step explicit: surprising observations motivate importing/constructing explanatory concepts that reorganize the case (Tavory & Timmermans, 2014).

**Hermeneutics and the productive role of interpretation.** Hermeneutic traditions emphasize that interpretation can be "productive," generating new understanding by applying horizons and conceptual resources that exceed what is explicitly said (Gadamer, 1989; Ricoeur, 1976). This supports treating L4 as a qualitatively distinct epistemic move: meaning is constructed *through* an external lens.

For computational readers, L4 is the regime where the output's validity depends on whether the imported framework is appropriate and correctly applied (e.g., mapping a case to an established theoretical model, typology, or mechanism vocabulary). This is also the level most sensitive to out-of-context hallucination: an LLM can easily "sound theoretical" by naming frameworks without the warrants that a human theorist would provide.

### D.5. Boundary tests and operational heuristics

Because the boundary between L3 and L4 is the most failure-prone in both human annotation and LLM automation, we adopt explicit decision rules grounded in prior distinctions (manifest/latent; emic/etic; first-/second-order coding):

**L1 vs. L2 (restatement vs. organization).**   If the output could be produced by rephrasing or extracting without introducing a reusable label system, it is L1. If it introduces categories/themes that summarize recurring elements across instances, it is L2 (Braun & Clarke, 2006a; Krippendorff, 2018).

**L2 vs. L3 (topics vs. implicit commitments).**   If the output only says *what kinds of things* appear (themes/topics) it is L2. If it asserts *what must be true to make the case coherent*—e.g., inferred goals, norms, implied obligations, latent tensions, unspoken constraints—it is L3, supported by pragmatic and frame-based inference (Grice, 1975; Fillmore, 1982; Graneheim et al., 2017).

**L3 vs. L4 (within-frame inference vs. theoretical reframing).**   A practical discriminator is an **external-lens test**:

> If a competent reader could, using the situation's ordinary frame and shared commonsense/pragmatic knowledge, infer the meaning without importing a specialized construct, it is L3. If the meaning depends on introducing an external framework/etic vocabulary (and the output would be incomplete or uninterpretable without that framework), it is L4.

This test is consistent with emic/etic distinctions (Pike, 1954) and with qualitative guidance that theory-driven (directed) analysis differs from inductive/latent interpretation (Hsieh & Shannon, 2005; Gioia et al., 2013).

**Reliability and transparency.**    As semantic depth and conceptual scope increase, reliability is harder to achieve; qualitative methodology therefore emphasizes explicit code definitions, auditability of inference, and trustworthiness criteria (Lincoln & Guba, 1985; Graneheim & Lundman, 2004; Krippendorff, 2018). In our annotation protocol (Sec. 4), we operationalize this by requiring short rationales and by calibrating annotators on boundary cases, especially L3/L4.

**Why this matters for LLM-driven qualitative research.**    Current LLM tooling strongly supports L1–L2 outputs (summaries, clustering, thematic labels), but bridging to L3–L4 requires controlled inference: (i) making implicit meaning explicit without drifting beyond what the frame supports (L3), and (ii) applying external theories in a disciplined way, with warrants and limits (L4). Our four-level meaning-making axis isolates these distinct computational challenges, enabling more precise evaluation of "how far" an automated qualitative pipeline actually goes.

# E. Expanded Level of Modeling

**What we mean by "modeling".**    In this paper, *modeling* refers to the **explicit representational structure** a study produces to describe a phenomenon—what the units are, what relations hold among them, and what kinds of reasoning the representation is intended to support. This is closer to the notion of *models as representations* in system science and simulation (rather than "models" as parameterized predictors in ML) (Box, 1976; Zeigler et al., 2000; Law, 2015). Importantly, a modeling level is **orthogonal** to a meaning-making level: the same interpretive claim can be expressed as (i) a static thematic map, (ii) a phase timeline, (iii) a causal mechanism graph, or (iv) a dynamical feedback system.

We operationalize **four levels of modeling** that recur across qualitative research and sociotechnical-system analysis. The key discriminators are: (i) **time** (absent vs. present), (ii) **causality/mechanism** (absent vs. present), and (iii) **dynamical semantics** (static dependencies vs. iterative state updates with feedback).

## E.1. The four levels

**Level 1: Static configuration and relational models (themes as structure).**    Level 1 models are **static** representations of a system's structure. The typical output is a set of **themes / constructs / entities** plus **relationships** that are *not* interpreted as temporal evolution or causal production. This includes (a) thematic maps and thematic networks that systematize how themes relate in a corpus (Braun & Clarke, 2006a; Attride-Stirling, 2001), (b) social network / relational representations (ties among actors, groups, or organizations) used as a cross-sectional structure (Wasserman & Faust, 1994; Borgatti et al., 2009), and (c) static knowledge/configuration layouts such as concept maps used to organize concepts and relations without committing to temporal or causal semantics (Trochim, 1989; Novak & Cañas, 2008).

**Signature:** "what relates to what" (clusters, centrality, co-occurrence, association, configuration) in a snapshot. Relations may be labeled and even directed (e.g., asymmetric ties), but the representation does not claim that the system *evolves* or that A *causes* B. For CS/NLP readers, Level 1 is most naturally viewed as *graph extraction / clustering / structural summarization* over qualitative material.

**Level 2: Stage / phase / timeline models (time without causality).**    Level 2 models introduce **time** as a first-class organizing principle, but they still **do not** make the core representational commitment of causal/mechanistic production. The output is a **sequence** of stages/phases (or a timeline of events/episodes), often produced via temporal bracketing, narrative sequencing, or process description (Van de Ven & Poole, 1995; Langley, 1999; Pettigrew, 1990). A common form is a phase model of change (Phase 1 → Phase 2 → Phase 3) or a chronicle of key events, where the edges mean "next / then / later" rather than "produces / changes". Sequence-analytic perspectives also fall naturally here when the modeling product is primarily an over-time patterning of states/events rather than a causal mechanism model (Abbott, 1995; 2001).

**Signature:** "what happens when" (ordering, phases, episodes), with descriptive transition language. For CS/NLP readers, Level 2 corresponds to *segmentation and sequencing* problems (phase detection, timeline construction, episode modeling), where the representation encodes temporal order but not causal effect.

**Level 3: Causal dependency and mechanism models (directed influence).**    Level 3 models explicitly represent **causal or mechanism-like dependencies**. Nodes are constructs/states/actors, and edges indicate that one element *changes/enables/blocks* another, potentially with conditions, mediators, or moderators. This aligns with causal-graph traditions in social science and AI (Pearl, 2009; Spirtes et al., 2000), as well as qualitative/soft-systems artifacts that are intended as causal mechanism diagrams rather than mere association maps. Related representational families include

cognitive/causal maps and fuzzy cognitive maps when used as explicit influence structures (Kosko, 1986).

**Signature:** "why / how" in the sense of directed production (A → B as influence), but still often **read as a static dependency structure** rather than an iterated dynamical system. The model may be qualitative and non-executable; what matters is the directional, mechanism-committing semantics.

**Level 4: Dynamical systems / feedback / complex systems models (state + update + iteration).** Level 4 models treat the phenomenon as an **evolving system** whose behavior unfolds through **iterative state change**. The first-class units are **system components** (institutions, infrastructures, policy regimes, collective behaviors, capacities, constraints, resources) that have **states** which update over time (quantitatively or qualitatively). The model specifies **transactions** (mechanism-like interactions) that update component states, and it uses **feedback**, **delays**, and often **nonlinear responses** to explain system-level patterns over time (waves, escalation/relaxation cycles, overshoot and correction, lock-in, tipping points, collapse/stabilization) (Forrester, 1961; Sterman, 2000; Meadows, 2008).

Crucially, Level 4 is **not restricted** to classical stock–flow diagrams. A Level 4 model may be: (i) qualitative system dynamics / feedback-system reasoning (reinforcing/balancing loops as explanatory machinery); (ii) coupled regime/phase systems where components move among qualitative states and co-evolve; or (iii) executable simulation models (agent-based, discrete-event) when the simulation is used to explain emergent trajectories (Bonabeau, 2002; Macy & Willer, 2002; Epstein, 2006; Gilbert & Troitzsch, 2005; Law, 2015; Zeigler et al., 2000).

**Signature:** the paper's core model claim depends on **endogenous time evolution** (state at $t$ shapes state at $t+1$) and uses feedback/iteration as the explanatory engine, not merely as metaphor.

### E.2. Operational boundaries and guardrails

**Level 1 vs. Level 2: static structure vs. explicit temporality.** A paper is Level 1 if the primary model is a *snapshot* of themes/entities and their relations (association/ties/configuration) without encoding ordering, phases, or evolution. It is Level 2 when the model explicitly organizes the phenomenon as *over-time* stages, phases, or timelines (Langley, 1999; Pettigrew, 1990).

**Level 2 vs. Level 3: temporal ordering vs. causal production.** A paper remains Level 2 if arrows mean "next" (temporal succession) but the analysis does not commit to edges as causal/mechanistic ("produces/changes/enables"). It becomes Level 3 when directed relations are used to explain how one construct generates changes in another, i.e., when the representation has causal semantics (Pearl, 2009; Spirtes et al., 2000).

**Level 3 vs. Level 4: static causal graphs vs. dynamical system semantics.** Level 3 models can include cycles, and they can talk about "feedback" rhetorically. Level 4 requires something stronger: the paper must articulate **state + update semantics + iteration** so that the explanation of observed/predicted behavior depends on system evolution over time (often via feedback and delays) (Sterman, 2000; Forrester, 1961).

A practical diagnostic:

- If you can remove time iteration and the core model claim still stands (it remains a story about dependencies or pathways), it is likely Level 3.

- If the core claim depends on iterative updating to produce macro behavior patterns (waves, oscillation, overshoot, collapse, lock-in), it is Level 4.

**Why this matters for LLM-based qualitative automation.** This modeling axis makes concrete what is being produced and therefore what should be evaluated. Many current LLM pipelines naturally yield Level 1 products (themes + static relational maps) or Level 2 products (phase/timeline summaries). Moving to Level 3 requires reliable causal semantics and defensible mechanism claims; moving to Level 4 requires explicit state/update/feedback commitments that support trajectory-level explanations and intervention reasoning (Pearl, 2009; Sterman, 2000; Zeigler et al., 2000).

# F. Annotation Study Examples

## F.1. Dimension 1: level of Meaning-Making

F.1.1. **M1**: Descriptive (manifest, surface-faithful).

- (**Keith et al., 2017**):

  *"...and extract entity mentions. Mentions are token spans that (1) were identified as "persons" by spaCy's named entity recognizer, and (2) have a (firstname,lastname) pair as analyzed by the HAPNIS rule-based name parser,6 which extracts, for example, (John, Doe) from the string Mr. John A. Doe Jr.. To prepare sentence text for modeling, our pre-processor collapses the candidate mention span to a special TARGET symbol. To prevent overfitting, other person names are mapped to a different PERSON symbol; e.g. 'TARGET was killed in an encounter with police officer PERSON.'"*

  This paper proposes a framework of identifying "civilians killed by police" via "event-entity extraction". The framework is a mixture of Quantitative Analysis and Qualitative Analysis. However, Qualitative Analysis process only happens for event and entity extraction given unstructured data (news articles), serving as a preprocessing step for the proceeding Quantitative Analysis step. Hence, this is an example of **M1**.

- (**Nasar et al., 2021**):

  *"Non-neural Approaches. Widely used supervised approaches for NER include HMM, MEMM, SVM, and CRF. Traditional approaches relied solely on underlying algorithm and initial training data."*

  This paper discusses different algorithms for Named Entity Recognition (NER) tasks, which involve learning underlying patterns of entities for extraction. Despite the requirement for global knowledge aggregation to form generalized understandings of entity patterns, such knowledge relies heavily on statistical distributions rather than semantic meaning. For example, if an entity is replaced with an arbitrary symbol (e.g., A) or masked, the overall abstracted knowledge remains unchanged because it is based on distributional statistics. However, this invariance does not hold for semantic knowledge: replacing the word "Mars" with "Atlantic Ocean" in a context of exploration changes the content, even though both are entities. Such substitutions substantially alter higher-level abstracted themes, shifting from "Space Exploration" in the case of Mars to "Treasure Hunting" in the latter case. Because no semantic abstraction is performed, this paper exemplifies **M1** meaning-making.

Below are a few more sample papers with **M1** complexity: (Li et al., 2018; Zhong et al., 2023; Bekoulis et al., 2020; Cheng et al., 2024; Zhou et al., 2016)

F.1.2. **M2**: Categorical (themes, topics, patterned organization).

- (**Cemri et al., 2025a**):

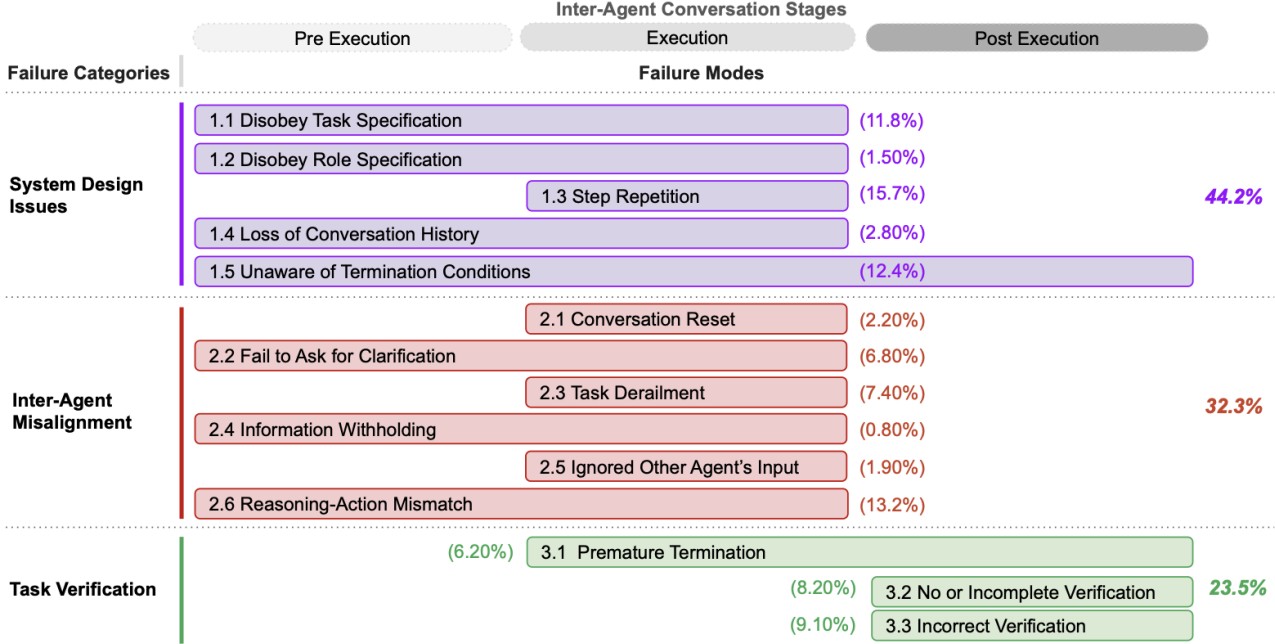

Figure 1: MAST: A Taxonomy of MAS Failure Modes. The inter-agent conversation stages indicate when a failure typically occurs within the end-to-end MAS execution pipeline. A failure mode spanning multiple stages signifies that the underlying issue can manifest or have implications across these different phases of operation. The percentages shown represent the prevalence of each failure mode and category as observed in our analysis of 1642 MAS execution traces. Detailed definitions for each failure mode and illustrative examples are available in Appendix A.

This paper proposes some of the failures often encountered in Multi-Agent Systems (MAS), which were uncovered from QA on MAS run trajectories that failed to complete the intended tasks. These themes are the result of observing MAS trajectories and synthesizing general patterns to describe the common failures of MAS. Hence, this work exemplifies **M2**, as it involves global semantic knowledge aggregation across instances, but does not introduce interpretive explanations or underlying assumptions beyond those directly supported by the observed data.

- (Wang et al., 2025):
  - *Parenting in the Digital Age*: Exploring the impact of digital technology on parenting practices and child development
  - *Work-Life Balance and Parenting*: Investigating the challenges faced by working parents and effective policies for achieving a healthy work-life balance.
  - *Mental Health Support for Parents*: Examining the importance of mental health resources for parents and strategies to improve access and support.

Similar to the MAS example, the high-level themes in this paper are also high-level, abstracted descriptions of what is observable in the dataset. Hence, this work is **M2**.

Below are a few more sample papers with **M2** complexity: (Fang et al., 2025; Li et al., 2022b; Barany et al., 2024; Edge et al., 2024; Zhong et al., 2023)

F.1.3. **M3**: INTERPRETIVE (IMPLICIT MEANING ENTAILED BY THE FRAME).

- (Fletcher & Sarkar, 2012):

    *Positive personality* Olympic gold medalists possessed numerous positive person- ality characteristics, such as openness to new experiences, consci- entiousness, innovative, extraverted, emotionally stable, optimistic, and proactive, which influence the mechanisms of challenge appraisal and meta-cognition. The following

*quote illustrates how one champion evaluated missing out on selection for a major international competition in a positive manner, due to his opti- mistic and proactive nature: There were four of us challenging for these final two places . and I got told I was on the reserve list. And at the time it was devastating but it's one of those things; if you don't take a ticket in the raffle, you're never going to win a prize. So you have to take the ticket . that's part of life and it just makes you think 'well, what can I do differently to make sure I do get success'?*

This paper explores the common characteristics of being an "Olympic champion" through interviews with Olympic athletes. The interview data consist of narratives about the athletes' daily lives and training practices. Themes such as "Positive Personality" cannot be derived through simple high-level textual summarization; instead, they require interpretive analysis of the interview material to address more complex questions such as "How is this useful?" or "What mindset characterizes an Olympic champion?" Because generating these higher level themes involves interpretive reasoning beyond descriptive aggregation, this work exemplifies **M3**.

- (**Rao et al., 2024**):

  - *Enhancing Transparency and Explainability*: Drivers are concerned about opaque fare calculations, unclear incentives, and uncertain criteria for earnings, surge pricing, and cancellations...

  - *Predictability and Worker Agency*: Drivers face unpredictable earnings from fluctuating surge pricing, algorithm changes, increased competition, complex incentive qualifications, and low compensation for long pickups and waits...

  - *Better Safety and More Time*: Drivers face navigation issues, support access difficulties, challenges with false complaints, and low compensation for additional tasks and wait times...

The data used in this paper were collected from Reddit forums and consist mainly of conversational snippets on diverse topics. One example from the data is: "I don't understand what the hell Uber is thinking when giving us long pickups for short trips. NOBODY sane accepts a \$4 ride for someone half an hour away." Mere descriptive abstraction would synthesize themes such as "User confusion with software algorithms," which focus on what is being said rather than why it is being said. To further understand the underlying message conveyed in the quotation and synthesize meaningful themes such as "Predictability and Worker Agency" or "Better Safety and More Time," interpretive analysis is required. Because such themes would not be possible without subjective interpretation, this paper provides a clear example of **M3**.

Below are a few more sample papers with **M3** complexity: (Hou et al., 2024; Montes et al., 2025; McKeown et al., 2015; Goulet et al., 2023; Nyaaba et al., 2025)

F.1.4. **M4**: Theoretical (external reframing; etic con-358 structs).

- (**Carius & Teixeira, 2024**):

  - *Health Barriers*: In addition to facing financial hardships, participants in our study expressed how these barriers affected their physical and mental well-being. Insufficient access to medical care further compounded their health concerns, making it challenging for them to achieve stability while dealing with mental and physical health issues. Many mentioned the absence of medical insurance or coverage, particularly among Black citizens...

  - *Insufficient Community Resources*: In interviews, participants also highlighted the significant challenges they face due to the lack of appropriate resources and infrastructure in their communities. They expressed concerns about the safety of their neighborhoods, the scarcity of well-paying job opportunities, and the need for more activities and community organizing efforts to improve their overall well-being...

  - *Public Assistance Barriers*: As we alluded to in earlier themes, participants in our study consistently emphasized the challenges they face with public assistance programs and the impact these barriers have on their journey toward financial stability...

This work clearly differentiates between **M3** and **M4**. While the themes are synthesized through interpretive analysis, the study also incorporates meta-contextual knowledge and draws on multiple domains to fully understand the operational mechanisms of a meta-system. For example, the work discusses the health barriers faced by many financially insufficient individuals, which prevent them from achieving financial stability. Furthermore, one contributing factor to this financial

instability is the lack of infrastructure investment needed to attract businesses and provide economic opportunities. Thus, understanding the operation and impact of the meta-system requires recognizing the interconnectedness of multiple domains, including racial barriers, infrastructure development, and socioeconomic conditions. Consequently, this work serves as a strong example of **M4**.

- (Tian et al., 2019):

    *As a result, seven main categories of factors were identified to influence Shaanxi Blower's business model innovation. These are market pressure, entrepreneurship, culture and strategy, technology, human resources, organizational capability and government policy. Market pressure and government policy are the direct external factors that promote business model innovation. Entrepreneurship is the direct internal factor that promotes business model innovation. Information technology and technological innovation, as the two technology-driven factors, directly affect business model innovation, but belong to different variables. Information technology together with government policy behavior and market pressure are exogenous variables, and technological innovation together with entrepreneurial spirit are endogenous variables. However, culture and strategy, human resources and organizational capability are the guarantee factors for business model innovation.*

    Similar to the discussion in the first example, understanding the "business more" system in this paper requires analyzing the interconnections among "market pressure, entrepreneurship, culture and strategy, technology, human resources, organizational capability, and government policy". Each of these domains is complex on its own, and together they form a cohesive, well-functioning system whose understanding requires **M4**-level analysis.

Below are a few more sample papers with **M4** complexity: (Binder & Edwards, 2010; Sun et al., 2020; Beattie et al., 2004; Barke et al., 2022; Rempel et al., 2013)

**F.2. Dimension 2: Level of Modeling**

F.2.1. **D1**: STATIC TAXONOMY AND RELATIONAL CONFIGURATION MODELS.

- (Cemri et al., 2025a):

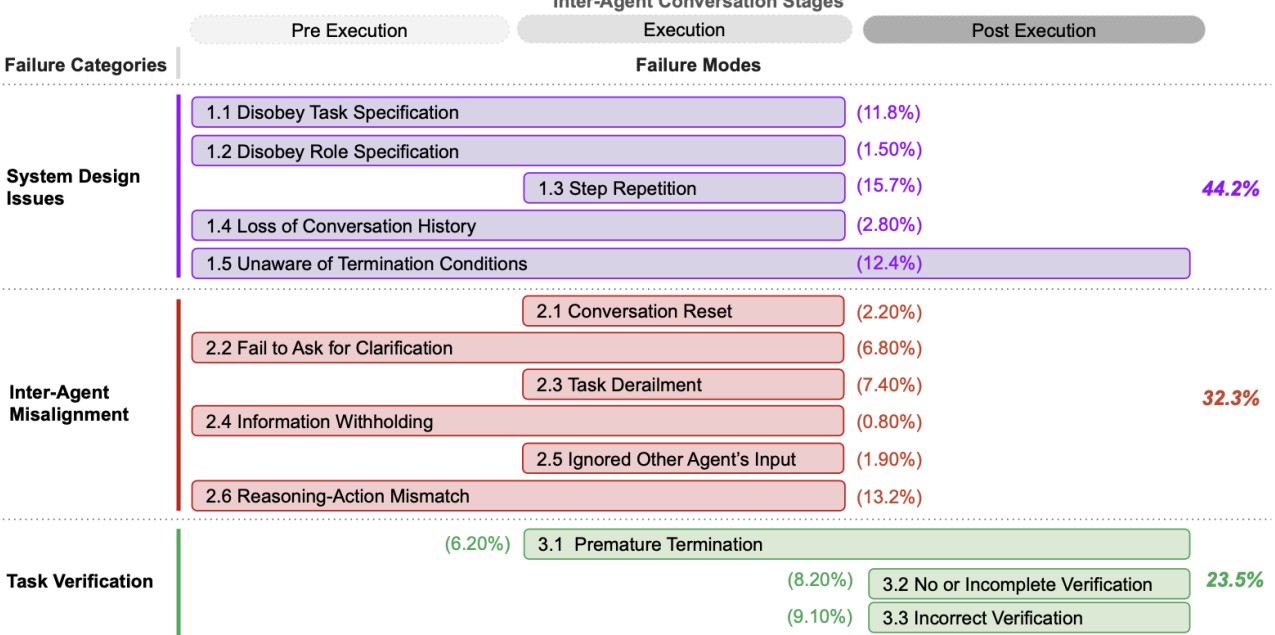

Figure 1: MAST: A Taxonomy of MAS Failure Modes. The inter-agent conversation stages indicate when a failure typically occurs within the end-to-end MAS execution pipeline. A failure mode spanning multiple

stages signifies that the underlying issue can manifest or have implications across these different phases of operation. The percentages shown represent the prevalence of each failure mode and category as observed in our analysis of 1642 MAS execution traces. Detailed definitions for each failure mode and illustrative examples are available in Appendix A.

The discovered themes are categorized into three main categories ("System Design Issues," "Inter-Agent Misalignment," and "Task Verification") as specified in the paper. Since this constitutes a hierarchical organization of themes and patterns, this is an example of **D1**.

- (**Zhong et al., 2025**):

| Communication and Engagement | Community and Social Responsibility | Crisis Management and Response |
|---|---|---|
| pain relief with addiction prevention | patient education focus | anticipating lost revenue |
| educating outlier prescribers | breaking prescription cycle | invalidating patents |
| focusing on cancer pain | investor pressure tactics | emphasizing side effects |
| using authority to build trust | activism for cannabis access | exalgo focus |
| providing a demo or trial | target young adult misusers | hostile takeover attempt |
| patient prescription history review | advocating for access to treatments | combating pharmacy shopping |
| promoting established medical procedures | politicizing the opioid crisis | comparing opioid to h.i.v. epidemic |
| identifying patient type using playbook | engaging with healthcare professionals | identifying lost prescriber opportunity |
| using patient education materials | public frustration with prices | threatening to stop subsidies |
| providing product sales objectives | opioid withdrawal treatment | anticipating opposition to expansion |

Figure 3: Three themes with randomly selecting cluster labels generated by HICode over OIDA.

Similar to the discussion in the first example, the themes are organized in a hierarchical taxonomy, with one category being "Communication and Engagement".

Below are a few more sample papers with **D1** complexity: (Pecoraro & Uusitalo, 2014; Serkina & Logvinova, 2019; Jäppinen et al., 2022; Gao et al., 2025; Lam et al., 2024)

F.2.2. **D2**: STAGE/PHASE/TIMELINE MODELS (TIME WITHOUT CAUSALITY).

- (**Li et al., 2023**):

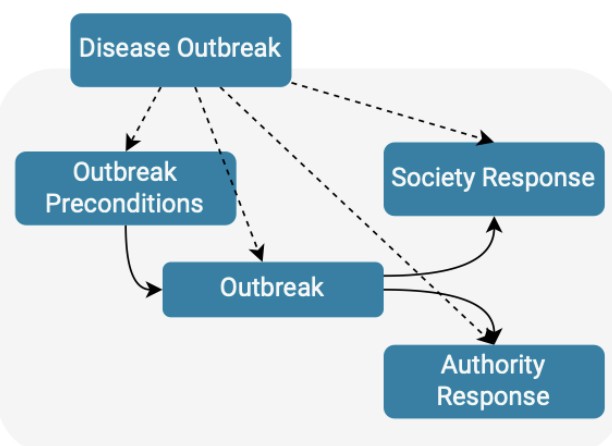

Figure 2: To create the schema for a given scenario, our model follows 3 rounds of operation: (1) event skeleton construction where we ask the LLM to list the important events; (2) event expansion to discover more related events for each existing event; event-event relation verification where we update the event-event relations based on the LLM's answers to questions about each event pair.

Figure 2 in this paper offers a glimpse of the output produced by the proposed QA framework, which organizes extracted events into a timeline with transition edges. This is an example of **D2**, as it incorporates a notion of time by ordering events in a sequential flow. However, it does not reach **D3**, because the relationships between events are temporal transitions rather than causal relationships.

- (**Li et al., 2020**):

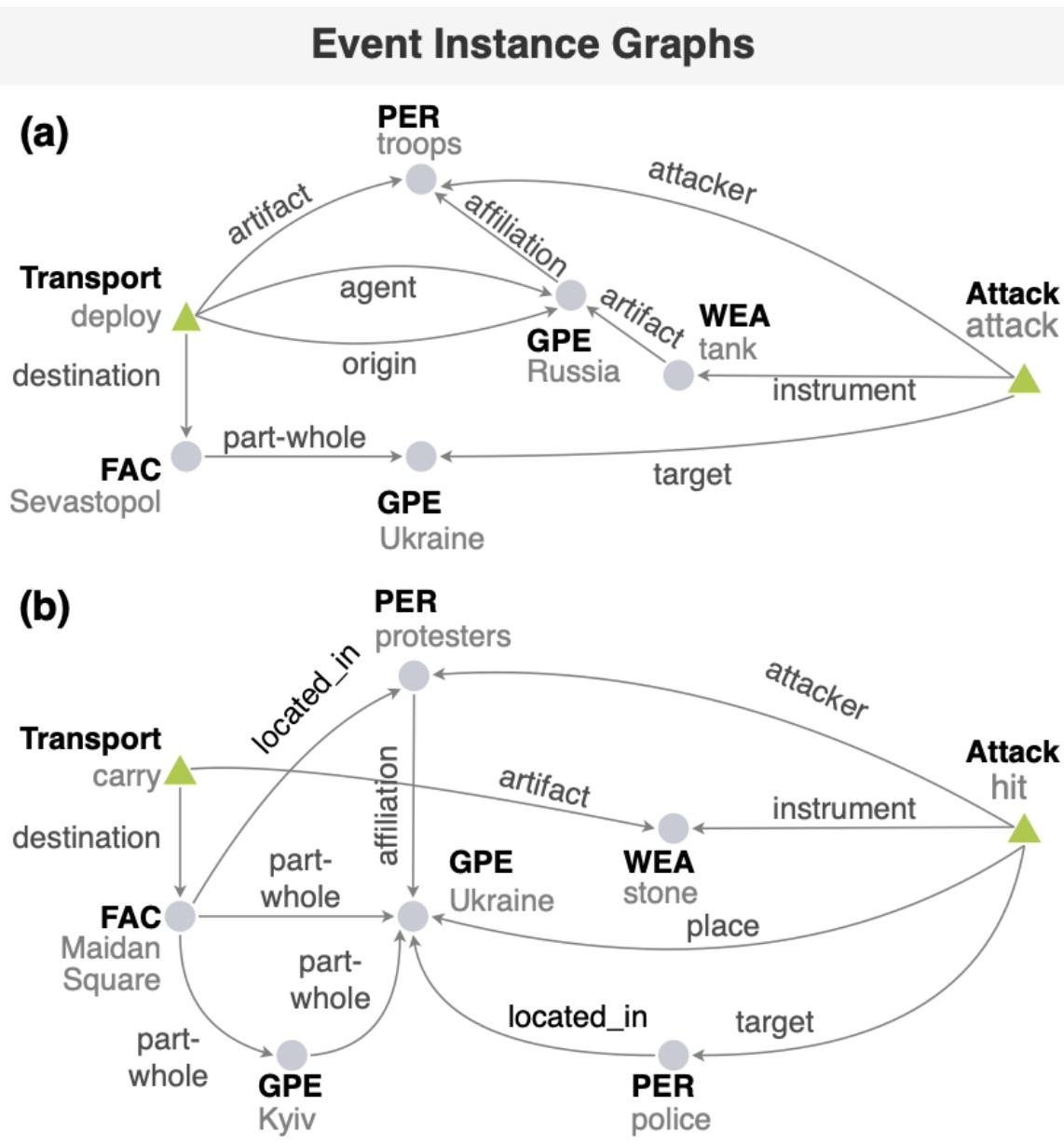

Figure 1: The framework of event graph schema induction. Given a news article, we construct an instance graph for every two event instances from information extraction (IE) results. In this example, instance graph (a) tells the story about Russia deploying troops to attack Ukraine using tanks from Russia; instance graph (b) is about Ukrainian protesters hit police using stones that are being carried to Maidan Square. We learn a path

language model to select salient and coherent paths between two event types and merge them into a graph schema. The graph schema between ATTACK and TRANSPORT is an example output containing the top 20

Figure 1 of this paper also provides a glimpse of the temporal structure of unfolding events, such as "Russia deploying troops to attack Ukraine" and "Ukrainian protesters hitting police." For each unfolding event, the temporal graph illustrates how the event progresses and how subevents transition from one to another. However, no causal relationships are specified between the events or subevents. Hence, this is an example of **D2**.

Below are a few more sample papers with **D2** complexity: (Lunney, 2025; Anggraini & Alim, 2023; Skryabina et al., 2016; Kristelstein-Hänninen, 2022; Tinder, 2022)

F.2.3. **D3**: CAUSAL DEPENDENCY AND MECHANISM MODELS (DIRECTED INFLUENCE).

- **(Du et al., 2022)**:

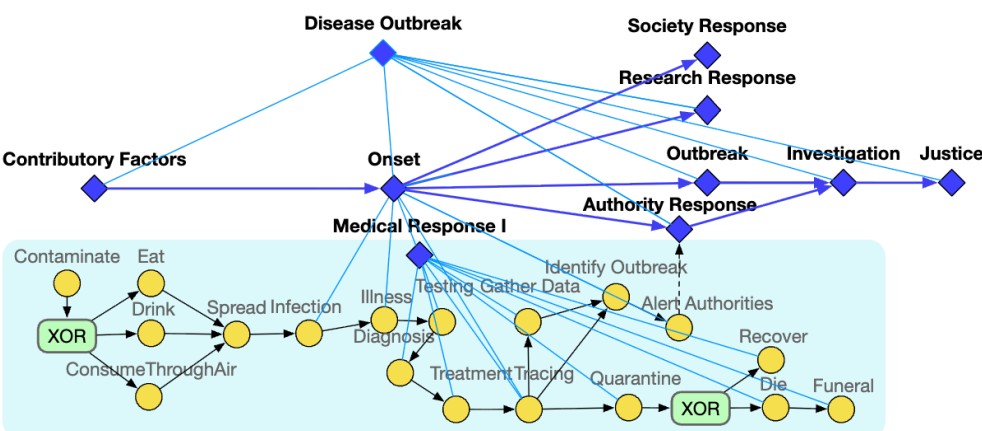

Figure 4: The curated schema for the disease outbreak scenario. Blue diamond shapes represent sub-schemas and yellow circles represent primitive events. Black arrows between primitive events represent temporal order, light blue lines between the primitive events and the sub-schema node represent event-subevent hierarchical relationship. Here we only show the primitive events under the Onset sub-schema

This paper is about event schema extraction and prediction. Its figure 4 shows one example of extracted schema, in which the events are connected by both temporal order (black line) and causal relationship (XOR operation). We view this as a partial causal graph and label this paper as **D3**.

- **(Safa et al., 2024)**:

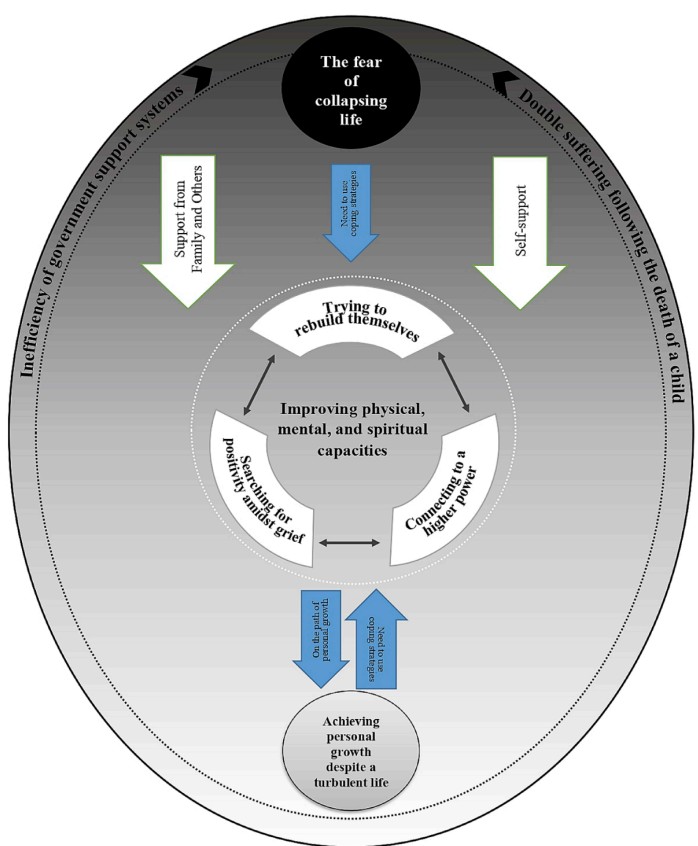

Figure 1 A schematic model "from fear of collapsing life to achieving personal growth"

This paper represents a directed pathway structure centered on how older adults move from 'fear of life collapse'. The model is explicitly structured around directed dependence. Figure 1 and the textual description in the 'Explanation of the model' section make clear that nodes represent factors/conditions/strategies (not persistent system components), and edges encode how elements contribute to, enable, or lead to others. The model's explanatory force comes from clarifying which factors contribute to the main concern, which strategies address it, and what facilitates movement toward the outcome-classic **D3** directed pathway logic, not **D4** loopy component interdependency.

Below are a few more sample papers with **D3** complexity: (Bratianu, 2020; Kendellen & Camiré, 2019; Agyeman, 2014; Attwell & Hannah, 2022; Chopra, 2019)

F.2.4. **D4**: YNAMICAL SYSTEMS / FEEDBACK / COMPLEX SYSTEMS MODELS (STATE + UPDATE + ITERATION).

- (**Kendellen & Camiré, 2019**):

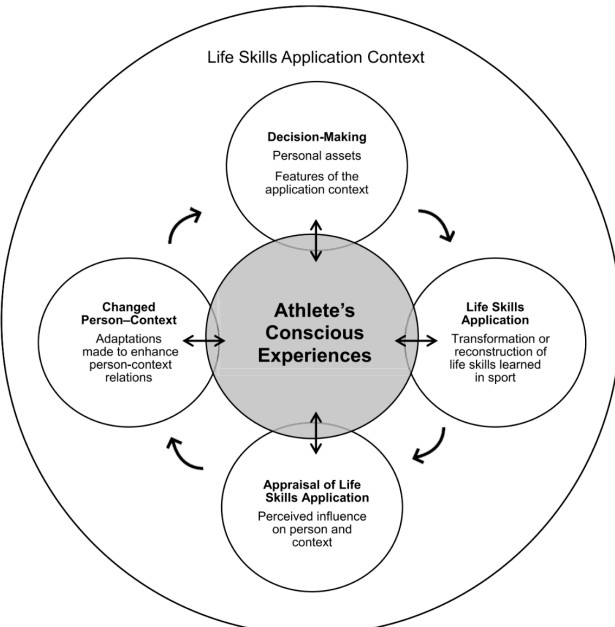

Figure 1. Grounded theory of life skills application.

The paper produces a cyclical, evolving system of person-context regulations. The authors define the process as having 'no definitive end point,' instead functioning as a continuous loop of decision-making, application, appraisal, and adaptation. The topology is defined by mutual coupling where the 'changed person-context' statefully feeds back into future decision-making cycles. Unlike a **D2** or **D3** model which would move from sport (source) to life (sink), this model treats the athlete and their environment as a dynamic system where knowledge and assets are continuously re-regulated across domains. So we annotate this as **D4**.

- (**Bratianu, 2020**)

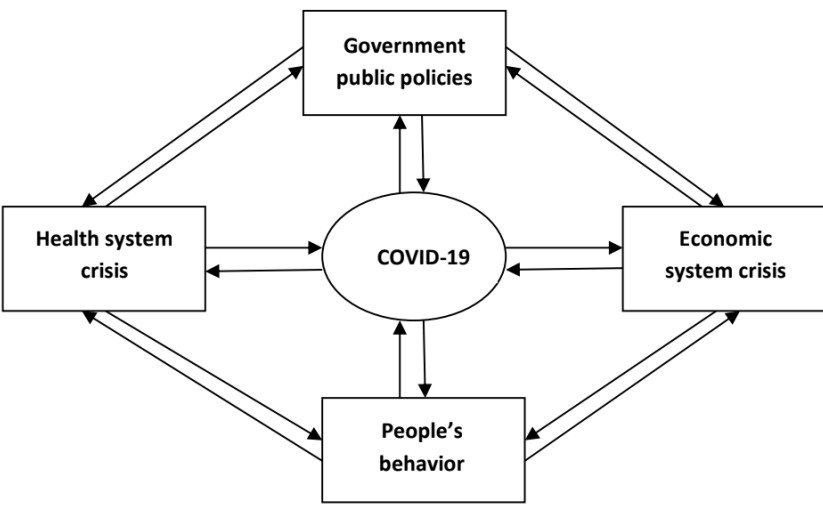

Figure 1. An integrated model of the COVID-19 pandemic dynamics concerning the health and economic systems Source: Author's own research

Based on the figure 1 of this paper, it explicitly defines the COVID-19 pandemic as a complex system of 'mutual coupling' and 'dense interdependencies' rather than a linear or directed sequence. The diagram shows correlations that 'act in all possible directions.' This violates the Source/Sink test (**D3**) because there is no start-to-end flow. Instead,

the model represents a state-based system where variables continuously influence and are influenced by the crises in a feedback web. The rationale for is rooted in this absence of a terminal state and the presence of iterative, stateful interdependency, which leads to **D4** annotation.

Below are a few more sample papers with **D4** complexity: (Karakas & Sarigollu, 2019; Coleman, 2017; Fellahi, 2021; Kendellen & Camiré, 2019; Loosemore, 1999)

# G. Annotation Study Details

## G.1. Research Paper Type Classification Prompt

```
You are a careful research assistant. Classify papers using only the provided fields. Do
↪  not invent details. If evidence is insufficient, choose Unclassified. Follow the label
↪  schema exactly and output valid JSON only.
User message template:
You will be given a academic research paper.
 Task: classify the paper into exactly one of these top-level categories:
Conceptual & Theoretical
 Empirical - xxx (xxx must be one of the empirical method labels listed below)
 Computational
 Review & Synthesis
 Quantitative
 Unclassified
Definitions and decision rules (apply in order)
A. Review & Synthesis
 Choose if the primary contribution is synthesizing prior literature rather than reporting
 ↪  new primary data. Strong cues: systematic review, scoping review, meta-synthesis,
 ↪  qualitative evidence synthesis, integrative review, realist review, narrative review
 ↪  (when it summarizes literature).
B. Computational
 Choose if the primary contribution is building/evaluating computational methods/tools to
 ↪  automate or computationally support qualitative research (e.g., NLP for coding,
 ↪  LLM-assisted thematic analysis, automated discourse analysis pipelines). It may
 ↪  include experiments, but the core is the computational framework/tool.
C. Quantitative
 Choose if the papers primary methodology emphasizes statistical analysis and modeling
 ↪  rather than the qualitative methods listed below. Strong cues include: regression
 ↪  analysis, causal inference, time series analysis, statistical modeling, hypothesis
 ↪  testing, significance tests (p-values), confidence intervals, effect sizes, large-N
 ↪  survey econometrics, predictive modeling as the main analysis, randomized experiments
 ↪  analyzed primarily quantitatively, etc.
 If the paper uses interviews/focus groups only as minor context but the main results are
 ↪  statistical/model-based, still choose Quantitative.
D. Empirical - xxx
 Choose if the paper reports collecting and analyzing real-world data about a phenomenon
 ↪  (participants, field/site, texts, artifacts, records), and qualitative analysis is
 ↪  central. Label must be one of:
Empirical - Interpretative Phenomenological Analysis
 Subcategories (pick one in rationale): lived experience of illness/care; professional
 ↪  identity/role; sensemaking/transition; trauma/grief; other.
 Cues: \IPA", \interpretative phenomenological", idiographic focus, small samples.
Empirical - Grounded Theory
 Subcategories: classic/Glaserian; Straussian; constructivist (Charmaz);
 ↪  other/unspecified.
 Cues: grounded theory, theory generation, constant comparative, theoretical sampling,
 ↪  saturation.
Empirical - Thematic Analysis
 Subcategories: reflexive TA; codebook TA; framework-informed TA; other/unspecified.
 Cues: \thematic analysis", \themes were identified", Braun & Clarke.
Empirical - Narrative Analysis
 Subcategories: life story/biographical; illness narratives; organizational narratives;
 ↪  other/unspecified.
 Cues: \narrative analysis", \stories", \plot", \narratives", temporality.
Empirical - Discourse Analysis
```

 Subcategories: conversation analysis; critical discourse analysis (CDA); discursive
 ↪ psychology; other/unspecified.
 Cues: discourse analysis, discursive, CDA, conversation analysis, turn-taking, rhetoric.
Empirical – Dialogue Analysis
 Subcategories: interactional/dialogic analysis; clinical dialogue; collaborative/team
 ↪ dialogue; other/unspecified.
 Cues: \dialogue analysis", \dialogic", \interaction episodes".
Empirical – Framework Analysis
 Subcategories: policy/service evaluation; health services; implementation/QI;
 ↪ other/unspecified.
 Cues: \framework analysis", matrix-based charting, applied policy research.
Empirical – Ethnography
 Subcategories: classic ethnography; focused ethnography; digital/virtual ethnography;
 ↪ institutional ethnography; other/unspecified.
 Cues: ethnography, participant observation, fieldnotes, prolonged engagement, site
 ↪ culture.
Empirical – Content Analysis
 Subcategories: qualitative content analysis; directed content analysis; summative content
 ↪ analysis; other/unspecified.
 Cues: content analysis, coding categories, frequency counts may appear but still
 ↪ qualitative.
Empirical – OTHERS
 Subcategories: case study (qualitative); phenomenology (non-IPA); interpretive
 ↪ description; template analysis; rapid qualitative analysis; participatory/action
 ↪ research; mixed-methods (qual-dominant); other.
 Use only if none of the above methods fit but it is clearly qualitative empirical work.
E. Conceptual & Theoretical
 Choose if the papers primary contribution is conceptual/philosophical/methodological
 ↪ discussion (arguments, frameworks, critiques) without primary data collection and not
 ↪ a review/synthesis.
F. Unclassified
 Choose if the provided fields are too sparse/ambiguous to justify A{E.
Output format (STRICT)
Return valid JSON only, with:
 rationale (100~200 words, grounded in evidence from the provided fields; mention what
 ↪ text triggered the decision; note uncertainties)
 decision (exact string from allowed labels)
Below is the paper you will classify:
========================

## G.2. Level of Meaning-Making Prompt

You are an expert qualitative researcher trained in multiple qualitative methodologies
↪ (e.g., Straussian grounded theory, constructivist grounded theory, thematic analysis,
↪ interpretative phenomenological analysis, discourse analysis, narrative analysis,
↪ content analysis). Your task is to evaluate a given qualitative research paper and
↪ judge the primary level of meaning-making at which the paper operates.
Critical clarifications (read carefully and apply strictly):
Methodology is not Level 4. The use of qualitative methods and coding procedures (e.g.,
↪ open/axial/selective coding, constant comparison, memoing, IPA steps, thematic
↪ analysis workflows, discourse-analytic techniques, codebooks, inter-coder reliability)
↪ does NOT by itself constitute Level 4 meaning-making. Level 4 is about the substance
↪ of the interpretive claims and what they depend on, not which qualitative methodology
↪ is used.
Theory citation is not Level 4 unless it is applied. Merely citing theories/frameworks as
↪ background, literature comparison, positioning, or discussion (e.g., introduction,
↪ related work, limitations, implications, future directions) does NOT qualify as Level
↪ 4. A paper qualifies as Level 4 only when external theory/frameworks/domain knowledge
↪ outside the corpus are actually used as interpretive machinery during coding,
↪ analysis, or theory development to generate or justify core findings.

Level 4 requires external interpretive resources beyond the corpus. To qualify as Level 4,
↪  the paper must rely on concepts, constructs, models, typologies, or explanatory
↪  systems that are not entailed by the corpuss semantic frame alone and that typically
↪  require domain expertise or specialized theoretical knowledge. These external
↪  resources must actively shape how codes/themes are constructed, linked, or elevated
↪  into the papers central claims.
Level 3 is not constrained to a fixed checklist. When judging Level 3, do not treat any
↪  list of sub-dimensions (e.g., teleology, emotion, causation, social meaning, norms,
↪  implicit assumptions, temporal structures,  pragmatics) as exhaustive. Level 3 can
↪  surface any kind of latent content that is implicitly entailed by the semantic frame
↪  of the data. The rule of thumb is: Level 3 uncovers non-trivial & interesting
↪  implicit/hidden meaning relevant to the research topic; Level 2 primarily condenses
↪  and organizes what is already explicit into categories.
Your goal is to infer what kind of meaning the paper primarily produces from its data:
↪  surface description, categorization, interpretive surfacing of latent meaning, or
↪  theory-based reframing using applied external frameworks.
Definitions of the four levels of meaning-making:
Level 1: Descriptive level of meaning-making
The paper primarily restates, summarizes, or reports what is explicitly present in the
↪  data. It stays close to participants words or observable events. Outputs are concrete
↪  descriptions of actions, experiences, events, and statements without abstraction
↪  beyond paraphrase or recounting. There is no systematic grouping into higher-level
↪  categories and no inference about implicit meaning. The central analytic move answers:
↪  what is said or observed?
Level 2: Categorical level of meaning-making
The paper primarily organizes descriptive observations into categories, themes, domains,
↪  or types. It condenses multiple surface details into higher-level labels that
↪  summarize recurring patterns. The categories remain largely faithful to the frame of
↪  the data and do not systematically surface latent motives, hidden meanings, implicit
↪  norms, emotions, assumptions, or causal structures. The central analytic move answers:
↪  what kind of phenomenon or recurring pattern is present?
Level 3: Interpretive level of meaning-making
The paper primarily brings forward implicit or latent meanings that are not explicitly
↪  stated in the data but are entailed by the semantic frame of the situations described.
↪  It uses reasoning and inference grounded in the text to articulate what is implicitly
↪  going on (e.g., goals/purposes, emotions, causal relations, social meaning, normative
↪  expectations, functional roles, assumptions, relational/coordination structure,
↪  pragmatic intent), but it is not limited to these; any latent content that is
↪  warranted by the datas frame may qualify. Level 3 does not depend on specialized
↪  external theories or domain-specific conceptual systems. The central analytic move
↪  answers: what is implicitly committed or implied by the data, beyond categorization?
Level 4: Theoretical level of meaning-making
The papers core analysis relies on external theories, frameworks, or domain-specific
↪  conceptual systems that are outside the corpus itself and not entailed by its semantic
↪  frame alone. These external interpretive resources are actively applied during coding,
↪  analysis, or theory development to generate, structure, or justify the main findings
↪  (e.g., coding through a theoretical lens, mapping data onto a formal model/typology,
↪  interpreting instances as manifestations of a named theoretical construct, using
↪  specialized constructs as the organizing backbone of themes). Crucially, merely
↪  mentioning or comparing to theory in the introduction/discussion does not qualify; the
↪  external framework must function as an analytic engine that shapes the results. The
↪  central analytic move answers: what broader conceptual system explains or re-situates
↪  the data as an instance of a larger theory?
Decision instructions:
A) Judge the primary level across the paper, not isolated passages. If multiple levels
↪  appear, choose the level that best characterizes the dominant analytic contribution
↪  (the main value add).
B) Do not classify as Level 4 simply because the paper uses a named qualitative
↪  methodology, uses the term theory, or includes theoretical citations.
C) Classify as Level 4 only if external theory/framework/domain knowledge outside the
↪  corpus is materially used to construct or justify codes, themes, relationships among
↪  themes, or the papers central explanatory claims.
D) If the paper mainly reports themes/categories without systematically surfacing latent
↪  meaning, prefer Level 2. If it systematically surfaces implicit meaning beyond
↪  categorization using reasoning grounded in the data, prefer Level 3.

E) In your rationale, cite concrete indicators from the papers analytic moves: how
↪  codes/themes are formed, what kinds of claims are made, and whether external theory is
↪  applied as an analytic mechanism versus merely cited for context.
Output format requirements:
Return valid JSON with exactly two fields and no additional text:
{
"judgement": "Level 1" or "Level 2" or "Level 3" or "Level 4",
"rationale": "Concise but specific justification grounded in the definitions above,
↪  explicitly distinguishing methodology-only references from applied external
↪  theoretical frameworks, and explaining why the papers primary meaning-making level is
↪  the chosen one."
}

## G.3. Level of Modeling Prompt

You are an expert qualitative researcher and methodologist familiar with how qualitative
↪  papers represent models (e.g., taxonomies/typologies, configuration/concept maps,
↪  stage/process models, causal graphs and DAG-style pathway models, workflows/finite
↪  state machines, and complex systems / systems-dynamics style models). Your task is to
↪  evaluate a given qualitative research paper and judge the primary level of modeling at
↪  which the paper operates.
Critical clarifications (read carefully and apply strictly):

Methodology is not a modeling level. The use of grounded theory, thematic analysis, IPA,
↪  discourse analysis, codebooks, inter-coder reliability, memoing, etc. does NOT
↪  determine the level of modeling. Judge the level only from the structure of the model
↪  the paper produces (figures, tables, or core result text).

A diagram is not required, but modeling commitments must be explicit. A paper may have no
↪  figure and still be any level if it clearly presents a taxonomy/configuration, a stage
↪  model, a directed dependence model (DAG/workflow), or a complex-systems
↪  interdependency model.

Terminology is not decisive; semantics are. Authors may use dynamic, system, model,
↪  framework, or causal loosely. Classify based on what nodes/edges mean and what the
↪  model is designed to explain, not the label the authors attach.
Loops alone are not sufficient for Level 4, but stateful stockflow is not required either.
↪  Level 4 is not restricted to numeric stocks/flows or continuous quantities. Level 4 is
↪  awarded when the papers model is best understood as a complex system of interacting
↪  components with rich interdependencies and transaction mechanisms, typically featuring
↪  multiple loops and no single end-to-end pathway structure.

Workflows and end-to-end pipelines are not Level 4 by default. A model can use system
↪  components/modules and still be Level 3 if it is essentially an inputoutput process
↪  with largely one-directional dependencies, clear start/end points, and limited
↪  feedback structure. Use the presence of clear sources/sinks and a dominant forward
↪  path as a strong Level 3 signal.

Source/sink test is a key discriminator. If the model has clear source nodes (starts) and
↪  sink nodes (ends)i.e., it can be read as a pipeline with inputs and outputsthat
↪  strongly indicates Level 3 (even if complex). Level 4 system interdependency models
↪  typically lack a single clear start or end and are organized around mutual coupling
↪  and loops.

Level 3 causal graphs are mostly DAG-like pathway models. Their primary purpose is usually
↪  to identify which factors/events contribute to which outcomes and to clarify
↪  pathways/mediators. Even if cycles appear, the models focus is not primarily
↪  feedback/interdependency structure of the system as a whole.

Level 4 models privilege system components and their interaction structure. Level 4 does
↪ not require explicit state-update semantics at the finest level; instead, it requires
↪ (i) persistent system components as the modeling units, (ii) concrete depictions of
↪ transactions/interactions among components, and (iii) a loopy/interdependent structure
↪ where feedback is central to how the system is represented (not merely an incidental
↪ cycle).
Your goal is to infer what kind of model the paper primarily produces: static structure,
↪ staged progression, directed pathway/transition logic (mostly DAG/workflow), or
↪ complex-systems interdependency modeling.

==============
Definitions of the four levels of modeling:
Level 1: Taxonomy and Configuration Models
The papers primary model is a static classificatory structure: a taxonomy/typology,
↪ descriptive conceptual map, component/configuration diagram, or topological
↪ organization. It organizes entities by hierarchy, grouping, partwhole structure, or
↪ simple association.
Level 1 models do not support analytic inference about relationships among categories
↪ beyond classification or co-presence. They do not systematically compare, align,
↪ contrast, or integrate categories across dimensions, cases, or levels in order to
↪ identify patterns, tensions, or higher-order relational structure.
The model does not represent temporal progression, staged movement, directed pathway
↪ logic, or system feedback/interdependency as its core. Connections indicate
↪ descriptive organization rather than analytic relations (e.g., dependency, opposition,
↪ alignment, or interaction).
The central modeling move answers: what exists, and how is it descriptively organized,
↪ rather than how elements relate analytically, change over time, or shape one another.
Level 2: Stage / Transition Models
The papers primary model is a discrete, ordered process described as movement through
↪ qualitatively distinct stages or phases (Step 1, Step 2, Step 3, ). Time is
↪ represented via explicit stage boundaries or milestones, and progression is defined in
↪ terms of entering, occupying, or exiting stages.
Stages are the representational primitive: they are treated as temporally ordered segments
↪ rather than analytically distinct social states or interacting components. Transitions
↪ primarily describe what typically happens next, not why movement occurs under specific
↪ conditions.
The model may be linear or may include relapse, looping, or return to earlier stages, but
↪ such recurrence is descriptive rather than explanatory. Movement between stages is not
↪ modeled as conditional on structured dependencies, mechanisms, or interacting
↪ sub-processes.
The model does not specify directed causal pathways among multiple factors, nor does it
↪ articulate feedback mechanisms or mutual interdependence among system components.
The central modeling move answers: what are the stages, and how does an entity progress
↪ through them over time?
Level 3: Directed Pathway Models (Mostly DAG Causal Graphs and Workflows/FSMs)
The papers primary model commits to directed dependence in a way that reads as a pathway,
↪ pipeline, or transition-logic structure. Directionality is analytically meaningful:
↪ edges specify that certain factors, events, or states enable, constrain, mediate, or
↪ lead to others.
 Causal graphs (mostly DAG-like): nodes represent factors, events, or variables, and
↪ directed edges represent which elements contribute to which outcomes (A  B). Models
↪ commonly include branching pathways, mediators, and identifiable end outcomes. Even if
↪ limited cycles appear, the overall structure retains recognizable sources and sinks
↪ and is used primarily to clarify contribution pathways, not to depict the system as
↪ mutually coupled.
 Workflows / FSMs: nodes represent steps, states, or modules, and edges represent
↪ conditional or sequential dependencies (e.g., if/then logic, decision points). The
↪ structure typically has clear inputs/starts and outputs/ends (or a dominant forward
↪ flow). Loops usually indicate rework, retry, or iteration within a pathway, rather
↪ than feedback that reorganizes the system as a whole.
Directed dependencenot temporal ordering aloneis the representational primitive. While
↪ time or sequence may be involved, the models analytic force comes from specifying
↪ which elements depend on which others, and under what conditions.

The model does not treat persistent system components as mutually shaping one another
↪   through dense interdependence, nor does it center feedback as a primary explanatory
↪   device.
The central modeling move answers: what directed pathways, dependencies, or transition
↪   logic link inputs or contributing factors to outcomes?
Level 4: Complex Systems Interdependency Models (Loopy component coupling + transactions)
The papers primary model treats the phenomenon as a system of interacting components whose
↪   behavior arises from dense interdependencies and concrete interaction/transaction
↪   mechanisms, rather than from a single pathway linking sources to sinks. The model is
↪   organized around multiple feedback loops, not a dominant forward progression.
 Units are persistent system components (e.g., institutions, policies, infrastructures,
↪   capacities, collective behaviors, constraints, resources, modules) that endure and
↪   interact over time, rather than isolated events, episodic factors, or stages in a
↪   pipeline.
 Edges represent concrete interactions or transactions among components (e.g., mutual
↪   influence, constraint or enablement, coupling, co-determination, reinforcement or
↪   balancing effects, cross-impacts). These interactions are treated as explanatory, not
↪   merely illustrative.
 The structure is characteristically loopy and interdependent: there is no single
↪   privileged start or end, and feedback is structurally central to how the phenomenon is
↪   represented and explained, not an incidental add-on.
 System behavior is explained via internal coupling, such that changes in one component
↪   alter the conditions under which other components operate. The model aims to account
↪   for emergent properties (e.g., nonlinearity, amplification, cascading effects,
↪   stability, resilience, or breakdown), even if numeric stocks/flows or formal dynamics
↪   are not specified.
The model cannot be meaningfully reduced to a staged progression or directed pathway
↪   without losing its explanatory intent.
The central modeling move answers: how do interacting system components mutually shape one
↪   another through interdependencies and feedback structure?
Decision instructions:
A) Judge the dominant modeling contribution across the paper.
 Assess the papers primary substantive model of the phenomenon, not isolated figures,
  ↪   illustrative examples, or secondary discussions. If multiple model forms appear,
  ↪   classify according to the model that carries the main explanatory burden of the
  ↪   results.
B) Do not assign Level 4 based on language or surface cues.
 Terms such as dynamic, system, iterative, complex, or citations to systems thinking are
  ↪   not sufficient. A paper qualifies for Level 4 only if its core model explicitly
  ↪   represents:
persistent system components,
concrete interaction or transaction mechanisms among those components, and
multiple feedback loops that are central to explanation.
 A single loop, recursive step, or metaphorical system reference is insufficient.
C) Apply the source{sink test as a strong discriminator.
 If the model has identifiable starting conditions and end outcomesand can be read as a
  ↪   start-to-end pathway, pipeline, or transition logicstrongly prefer Level 3, even if
  ↪   the pathway is complex or includes limited cycles.
D) Prefer Level 4 only when mutual coupling is the primary representational logic.
 Assign Level 4 only when the model is best understood as a mutually coupled system:
no single dominant trajectory or privileged start/end point,
dense interdependencies rather than mostly one-directional influence, and
explicit depiction of interaction mechanisms that explain system behavior through
↪   feedback.
E) Ground the rationale in concrete representational features.
 In the justification, explicitly reference:
what nodes represent (categories vs stages vs events/variables vs persistent system
↪   components),
what edges encode (association vs temporal progression vs directed dependence vs
↪   interaction/transaction),
whether the structure is pathway-like with sources/sinks or system-like with pervasive
↪   coupling, and
whether feedback/interdependency is structurally central or merely incidental.

```
IMPORTANT (algorithm vs domain): Judge the level of modeling based on the papers
↪   substantive model of the target phenomenon, not the computational pipeline. Multi-step
↪   LLM workflows (generation  clustering  evaluation), iterative loops, or module
↪   diagrams describe the method, not the domain model. If the main output is
↪   themes/categories/hierarchies of labels, classify as Level 2 (or Level 1 if its mostly
↪   descriptive with no real categorization). Only assign Level 3/4 when the papers
↪   results explicitly model directed dependence (L3) or loopy component interdependencies
↪   (L4) in the real-world system being studied.
Output format requirements:
Return valid JSON with exactly two fields and no additional text:
{
"judgement": "Level 1" or "Level 2" or "Level 3" or "Level 4",
"rationale": "Concise but specific justification grounded in the definitions above,
↪   explicitly referencing the papers primary model form, what nodes/edges mean, whether
↪   the structure is sourcesink pathway-like (Level 3) versus loopy component-coupled
↪   system (Level 4), and why the chosen level best matches the dominant modeling
↪   contribution."
}
```

## H. Operationalizability Experiment Details

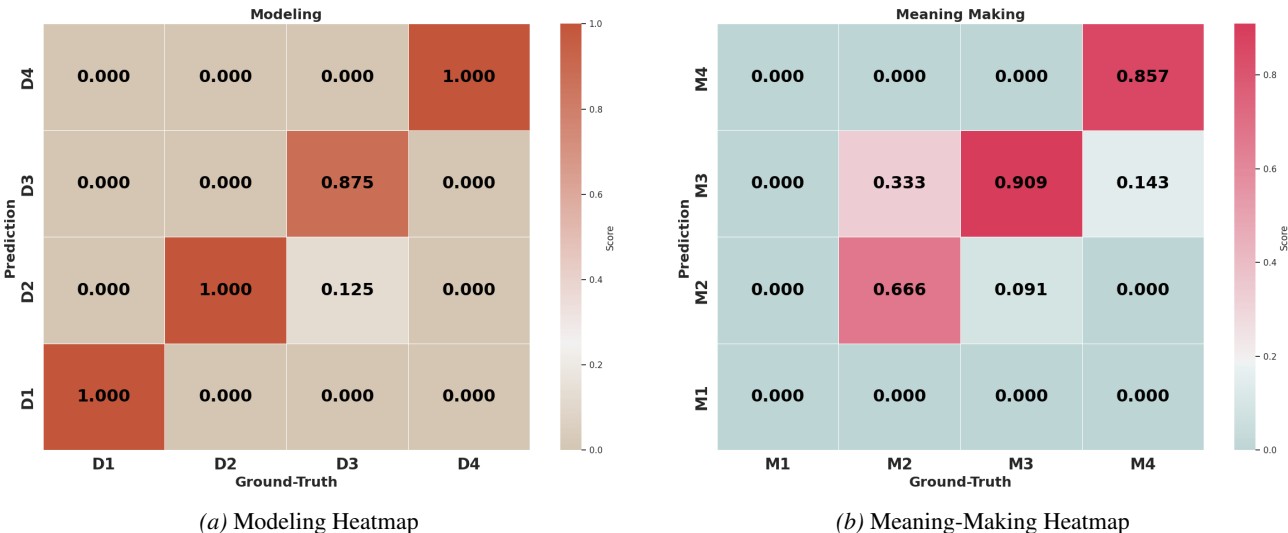

*(a)* Modeling Heatmap       *(b)* Meaning-Making Heatmap

*Figure 4.* Comparative Analysis of Qualitative Research Models

| Levels | # Papers | Levels | # Papers |
|--------|----------|--------|----------|
| M1 | 0 | D1 | 3 |
| M1 | 3 | D1 | 11 |
| M1 | 11 | D1 | 16 |
| M1 | 21 | D1 | 3 |

*Table 3.* Ground-truth distribution of sampled papers across Level of Meaning-Making and Level of Modeling
.

**Cohen's Kappa Scores:**

- Meaning-Making : **0.744**

- Modeling: **0.906**

We computed two confusion matrices comparing GPT-5.2 predictions (rows) with human ground-truth annotations (columns) on 35 sampled papers for both Level of Meaning-Making and Level of Modeling. To calculate Cohen's Kappa scores, the values in the confusion matrices were first normalized per ground-truth level to account for the uneven distribution of levels for both Level of Meaning-Making and Level of Modeling, and the resulting matrices were used for the computation.

# I. Operationalizing the Agenda: Protocols, Fit Metrics, Privacy, and Induction

This appendix provides concrete "first implementation paths" for several agenda items raised in Sec.5 — especially (i) moving from D1 coding outputs to D2–D4 structured models, and (ii) operationalizing "fitness-to-corpus" for D2/D3/D4 structures (e.g., evidence coverage, contradiction rate, semantic consistency) in a way that is auditable and automatable. These operationalizations are designed to be compatible with our modeling-level definitions (D2 time-without-causality; D3 directed causal pathways; D4 state+update+iteration/feedback) and with the paper's emphasis on traceability and data-governance constraints for qualitative corpora.

## I.1. Representational protocols: a typed, evidence-anchored graph standard for D2/D3/D4

**Goal.** We want one interoperable representation that (i) cleanly encodes the additional commitments as we move from D2→D3→D4, (ii) keeps a strict audit trail from every node/edge to supporting excerpts (mitigating loss-of-traceability risks), and (iii) can be serialized, validated, and scored by "fitness-to-corpus" metrics (next subsection).

**Core object: `QualGraph` (typed property multigraph).** Let a qualitative model be a directed, typed property multigraph:

$$G = (V, E), \quad v \in V \text{ nodes}, \quad e \in E \text{ edges.}$$

Each node/edge carries (a) a *semantic type*, (b) a *human-readable definition*, and (c) *evidence anchors* into the corpus.

**Evidence anchors (required for every node and edge).** We standardize an `EvidenceItem` that references an excerpt and records why it supports the claim.

```
EvidenceItem := {
  excerpt_id: str,            # stable ID
  doc_id: str,                # stable ID (may be hashed)
  span: [int, int],           # char offsets in excerpt text (or token offsets)
  time: Optional[TimeRef],    # if available; see \model{2}/\model{4}
  support_label: {SUPPORTS, CONTRADICTS, MENTIONS},
  rationale: str,             # short justification (human or model-generated)
  source: {HUMAN, MODEL, HYBRID},
  confidence: float           # [0,1], optional but recommended
}
```

This directly supports the "evidence-linked" requirement emphasized in the paper's discussion of rigor and trustworthiness.

**Node schema (shared across D2/D3/D4).**

```
Node := {
  id: str,
  node_type: str,             # e.g., STAGE, EVENT, CONSTRUCT, STATE_VAR, ACTOR, POLICY
  label: str,                 # short name
  definition: str,            # what counts as an instance of this node
  aliases: List[str],         # optional, for retrieval/normalization
  attributes: Dict,           # optional typed fields (below)
  evidence: List[EvidenceItem]
}
```

**Edge schema (shared across D2/D3/D4).**

```
Edge := {
  id: str,
```

```
  src: str, dst: str,        # node IDs
  edge_type: str,            # e.g., NEXT, ENABLES, INCREASES, DECREASES
  polarity: Optional[int],   # +1/-1 for signed influence; None otherwise
  qualifiers: Dict,          # conditions, moderators, delay, scope, etc.
  evidence: List[EvidenceItem]
}
```

**D2 protocol: stage/phase/timeline graphs (time without causality).**  D2 models make time first-class, but edges mean "next/then" rather than "produces/changes". We encode D2 with:

- **Node types:**

  - STAGE: a phase/episode with an operational definition and boundary cues.
  - EVENT: an atomic happening (optionally used; D2 may be stage-only).
  - MARKER (optional): boundary markers (e.g., "handoff", "policy change").

- **Edge types (temporal semantics only):**

  - NEXT(STAGE_i, STAGE_j): $i$ precedes $j$.
  - CONTAINS(STAGE, EVENT): event occurs within a stage.
  - OVERLAPS(STAGE_i, STAGE_j) (optional): permitted if the analysis allows co-occurring phases.

- **Required attributes:**

  - For each STAGE: boundary rules or cues (attributes.boundary_cues), e.g., lexical markers, role transitions, or criteria.
  - For each NEXT edge: evidence excerpts that (i) instantiate both stages and (ii) warrant ordering (timestamps, narrative order, "before/after" language).

**D3 protocol: causal pathway graphs (directed influence).**  D3 edges encode causal/mechanism-like dependencies (enable/constrain/produce/change/mediate). We encode D3 with:

- **Node types:** CONSTRUCT (factor/state/condition/strategy), ACTOR (optional), OUTCOME (optional).

- **Edge types (directional semantics):**

  - CAUSES, ENABLES, INHIBITS, CONSTRAINS, MEDIATES, MODERATES.

- **Required qualifiers on edges:**

  - qualifiers.mechanism_sketch: one-sentence "how" claim (even if partial).
  - qualifiers.conditions: when/for whom the edge holds (if present in the corpus).
  - polarity: optional sign for monotone effects (e.g., "more A tends to increase B").

- **Evidence requirement:** each edge must include (a) *support excerpts* and (b) at least one *searched-for counterexample* excerpt (even if labeled IRRELEVANT), to make contradiction-rate measurable (next subsection).

**D4 protocol: qualitative system dynamics graphs (state + update + iteration).**  D4 models treat the phenomenon as an evolving system whose behavior depends on iterative state change (state at $t$ shapes state at $t+1$), often with feedback and delays. We encode D4 with:

- **Node types:**

  - STATE_VAR: a system component with an interpretable state (capacity, resource, sentiment, constraint, norm, etc.).
  - FLOW (optional): if using stock–flow style; otherwise omit.
  - REGIME (optional): named qualitative regimes (e.g., "tight control", "relaxed control").

- **Edge types (influence + dynamics qualifiers):**

    - `INCREASES(src, dst)` with `polarity=+1`; `DECREASES` with `polarity=-1`.
    - `DELAYED_INFLUENCE`: same as above but with `qualifiers.delay_type` (e.g., "short/long", or "administrative delay").
    - `TRANSACTION`: concrete interaction updating states (used when the analysis specifies a mechanism-like transaction).

- **Loop objects (first-class, derived or explicit):**

```
Loop := {loop_id, node_ids: List[str], loop_sign: {REINFORCING,BALANCING,UNKNOWN},
         evidence: List[EvidenceItem]}
```

Loop sign can be computed from edge polarities when available; otherwise left `UNKNOWN` and evaluated via evidence coverage/closure (next subsection).

- **Minimal update semantics:** Each `STATE_VAR` has a discrete qualitative state $\in \{\text{LOW}, \text{MID}, \text{HIGH}\}$ (or ordinal bins). Each timestep applies signed influences:

$$x_j(t+1) = \text{clip}\Big(x_j(t) + \sum_i s_{ij} \cdot \Delta(x_i(t))\Big),$$

where $s_{ij} \in \{-1, +1\}$ is edge polarity and $\Delta(\cdot)$ maps ordinal state to a signed "pressure" (e.g., LOW=-1, MID=0, HIGH=+1). This is intentionally a toy formalization, but it is sufficient to produce directional predictions that can be compared to corpus-extracted trend statements.

**Serialization and validation.** A first implementation can store `QualGraph` as JSON (+ JSONSchema / Pydantic validation), and load into NetworkX for graph operations. Recommended minimum validators: (i) all nodes/edges have $\geq 1$ evidence item; (ii) **D2** `NEXT` edges form an acyclic order unless explicitly flagged as recurrent; (iii) **D4** edges have polarity if they are used in loop sign or simulation.

### I.2. Goodness-of-fit for qualitative models: evidence-anchored model checking with LLM judges

**Motivation.** The agenda calls for evaluation methodologies that operationalize "fitness-to-corpus" for **D2**/**D3**/**D4** (coverage, contradiction rate, pattern matching, semantic consistency), but such metrics are nonstandard partly because QR outputs are narrative and because closed corpora complicate benchmarking. Here we define a scoring pipeline that produces (a) element-level support/contradiction labels, and (b) a model-level scalar score with transparent components.

**Data model (corpus as testable datapoints).** Let the corpus be segmented into excerpts:

$$\mathcal{X} = \{x_1, \ldots, x_n\},$$
$$x := (\text{text, doc\_id, time, metadata}).$$

Time may be an absolute timestamp (e.g., log time), a relative narrative index, or a document-local order. For interviews without timestamps, `time` can be set to the turn index.

Let a qualitative model (**D2**/**D3**/**D4**) be a `QualGraph` $G$ with a set of *atomic claims*:

$$\mathcal{K}(G) = \mathcal{K}_{\text{node}} \cup \mathcal{K}_{\text{edge}} \cup \mathcal{K}_{\text{struct}},$$

where:

- $\mathcal{K}_{\text{node}}$: "Excerpt $x$ instantiates node $v$" (stage membership, construct presence, state-var mention).

- $\mathcal{K}_{\text{edge}}$: "Excerpt $x$ supports relation $e$" (NEXT, ENABLES, INCREASES, etc.).

- $\mathcal{K}_{\text{struct}}$: structure-dependent claims (e.g., **D2** order constraints; **D4** loop closure; **D4** trend predictions from qualitative simulation).

**LLM judge as a claim-labeling function (with auditability).** Define a judge function that takes a claim $k$ and an excerpt $x$ and returns:

$$J(k, x) \rightarrow (y, c, r),$$

where

$$y \in \{\text{SUPPORTS}, \text{CONTRADICTS}, \text{IRRELEVANT}\}$$

, $c \in [0, 1]$ is confidence, and $r$ is a short rationale. We store $(y, c, r)$ back into `EvidenceItem`. This explicitly addresses the "loss of traceability" risk if multi-step synthesis is not evidence-linked.

**Retrieval: choosing which excerpts to judge.** For each node/edge, retrieve candidate excerpts via a lexical+embedding query using `label`, `aliases`, and `definition`. A first implementation can use:

- BM25 over excerpts for lexical recall;

- dense embeddings for semantic recall;

- optional filters: same `doc_id`, time window, speaker role.

**Toy formalization: per-claim support and global fitness.** For any claim $k$, let $S_k$ be the set of retrieved excerpts judged SUPPORTS, and $C_k$ those judged CONTRADICTS. Define:

$$\text{score}(k) = \frac{|S_k| - \lambda |C_k|}{|S_k| + |C_k| + \epsilon},$$

with $\lambda > 1$ penalizing contradictions and $\epsilon$ preventing division by zero. Define *coverage*:

$$\text{cov}(k) = \mathbb{I}[|S_k| \geq m],$$

i.e., the claim is "covered" if at least $m$ supporting excerpts exist.

Then:

$$\text{Fit}(G, \mathcal{X}) = \underbrace{\frac{1}{|\mathcal{K}|} \sum_{k \in \mathcal{K}} \text{score}(k)}_{\text{support–contradiction}}$$

$$+ \beta \underbrace{\frac{1}{|\mathcal{K}|} \sum_{k \in \mathcal{K}} \text{cov}(k)}_{\text{evidence coverage}}$$

$$- \gamma \underbrace{\text{Complexity}(G)}_{\text{penalty}}.$$

where a simple complexity term is $\text{Complexity}(G) = |V| + |E|$ (or $|V| + |E| + |\text{Loops}|$ for **D4**). This yields an explicit lever for *adaptive level selection*: prefer the most committal model (**D4**¿**D3**¿**D2**) that wins under a complexity-penalized fit.

**D2-specific fit components (time without causality).** **D2**'s signature is ordering and segmentation. We add:

- **Stage assignment coverage:** fraction of excerpts assignable to at least one stage with high-confidence SUPPORTS.

- **Order consistency:** extract a per-document stage sequence $\pi_d$ by labeling excerpts with stages and collapsing consecutive repeats; compute violations of `NEXT` constraints.

- **Boundary clarity:** for each adjacent pair of stages, measure how often the judge can identify boundary cues (explicit markers, turning points) versus "ambiguous transition" labels.

**D3**-specific fit components (directed influence). **D3** edges require directional warrant and mechanism evidence. We add:

- **Directional support:** judge whether excerpts warrant $A \rightarrow B$ rather than $B \rightarrow A$.

- **Mechanism specificity:** judge whether the excerpt provides (even partial) "how" content matching `qualifiers.mechanism_sketch`.

- **Counterevidence search:** retrieve excerpts mentioning $A$ without $B$ (or vice versa) and allow the judge to label them as CONTRADICTS / IRRELEVANT—this operationalizes contradiction rate.

**D4**-specific fit components (state + update + iteration). **D4** requires evidence for feedback/iteration, not just rhetorical "systems" language. We add:

- **Loop closure:** for each loop, require that each constituent edge has $\geq m$ supports *and* at least one excerpt supports recurrence/cycling across time (e.g., "it comes back," "repeats," "oscillates," "tighten then relax").

- **Trend consistency via qualitative simulation:** run the toy ordinal update semantics to generate predicted directions (up/down/no-change) for key variables; extract trend statements from the corpus (LLM-based) and compute agreement.

**Pseudocode**

```
Algorithm 1: Evidence-Anchored Goodness-of-Fit (\model{2}/\model{3}/\model{4})

Inputs:
  X: list of excerpts x = (text, doc_id, time, metadata)
  G: QualGraph with Nodes V and Edges E (and optional Loops)
  k: number of retrieved excerpts per claim
  m: minimum supports for coverage
  Judge(): LLM judge returning (label, confidence, rationale)

Preprocess:
  Build retrieval index over X (BM25 + embeddings).
  Optionally normalize time fields (per-doc ordering if needed).

1) NODE CHECKING (coverage + support)
  For each node v in V:
    Qv := build_query(v.label, v.aliases, v.definition)
    Cands := RetrieveTopK(X, Qv, k)
    For each x in Cands:
      y,c,r := Judge(claim="x instantiates v", excerpt=x, node=v)
      Store EvidenceItem(x, y,c,r) on v
    node_score[v] := score(claim=v, from stored EvidenceItems)

2) EDGE CHECKING (support + contradiction)
  For each edge e=(u->v) in E:
    Qe := build_query(u, v, e.edge_type, e.qualifiers)
    Cands_sup := RetrieveTopK(X, Qe, k)
    Cands_ctr := RetrieveCounterevidence(X, u, v, e)   # e.g., u w/out v, reverse order
    For each x in Cands_sup U Cands_ctr:
      y,c,r := Judge(claim="edge e holds", excerpt=x, edge=e)
      Store EvidenceItem(x, y,c,r) on e
    edge_score[e] := score(claim=e, from stored EvidenceItems)

3) STRUCTURE CHECKING (level-specific)
  If G is \model{2}:
    sequences := ExtractStageSequences(X, V, Judge)  # label excerpts -> stages ->
    ↪  collapse
    \model{2}_struct_score := 1 - OrderViolationRate(sequences, NEXT edges)
  If G is \model{4}:
    loops := DetectOrReadLoops(G)
    \model{4}_loop_score := LoopClosureScore(loops, edge_score)
    sim_pred := QualitativeSimulate(G, T=some horizon)
```

```
    trend_obs := ExtractTrendsFromCorpus(X, STATE_VAR nodes, Judge)
    \model{4}_trend_score := TrendAgreement(sim_pred, trend_obs)

4) AGGREGATE
  Fit := mean(node_score) + mean(edge_score) + level_struct_score - gamma*Complexity(G)
  Return Fit and a diagnostic report (worst edges, contradictions, low-coverage nodes).
```

**Minimal judge templates (implementation hint).**   A workable first prompt pattern is:

- **Node claim:** "Given the node definition, does the excerpt instantiate it? Output {SUPPORTS/IRRELEVANT} and one sentence why."

- **Edge claim:** "Given edge type and direction (and polarity/conditions), does the excerpt support it, contradict it, or is it irrelevant? Output label + short rationale; do not infer beyond the excerpt."

Multiple independent judge samples (e.g., $n=3$) and majority vote can reduce variance; low-agreement cases become candidates for human adjudication, consistent with the paper's caution about hallucination and over-interpretation.

### I.3. Privacy-preserving "gold artifacts": de-identification pipelines and tooling

**Why this is not optional.**   The paper explicitly notes that primary qualitative materials and codebooks are often closed due to privacy/confidentiality constraints, limiting reuse and evaluation, and that using external model services raises data governance and confidentiality risks unless de-identification and secure deployment are in place. Therefore, any recommendation to release shareable gold artifacts must come with a concrete privacy path.

**What to share (pragmatic tiering).**   A realistic "shareable gold" bundle can be tiered:

- **Tier A (lowest risk):** codebook (definitions + boundary/counterexamples), model graphs (D2/D3/D4) with evidence pointers to excerpt IDs, and aggregate statistics (co-occurrence, precedence matrices). No raw text.

- **Tier B (moderate risk):** de-identified excerpts (redacted/pseudonymized) plus trace links to nodes/edges.

- **Tier C (high risk, controlled):** original raw transcripts under DUA / secure enclave; evaluation code runs "in place."

**De-identification operations (what to remove/transform).**   At minimum, target direct and quasi-identifiers:

- **Direct identifiers:** names, emails, phone numbers, IDs, usernames.

- **Quasi-identifiers:** specific dates, fine-grained locations, rare job titles, unique events.

Typical transformations:

- **Redaction:** remove span entirely (e.g., emails).

- **Pseudonymization:** replace with consistent placeholders (e.g., `[PERSON_7]`), preserving within-document coreference.

- **Generalization:** "March 12, 2023" $\rightarrow$ "March 2023" or "2023"; "Cambridge, MA" $\rightarrow$ "Northeast US".

- **Date shifting (optional):** add a random per-document offset while preserving intervals.

**Python tooling suggestions (drop-in building blocks).**   A first implementation can combine:

- `presidio-analyzer` + `presidio-anonymizer` (PII detection + replacement rules)

- `spacy` (NER; custom entity ruler for domain-specific identifiers)

- `stanza` (alternative NER)

- `scrubadub` (simple PII scrubbing for common patterns)

- `phonenumbers` (robust phone parsing), regex for IDs, `dateparser` for date normalization

- `rapidfuzz` (string similarity to cluster mentions for consistent pseudonyms)

De-identification is not a guarantee; high-risk releases should still use Tier C controls.

### Pseudocode

```
Algorithm 2: De-identify a qualitative corpus with consistent pseudonyms

Inputs:
  X: excerpts (text, doc_id, time, ...)
  pii_detectors: [Presidio, spaCy NER, regex detectors]
  policy: which entity types to redact vs pseudonymize vs generalize
Outputs:
  X': de-identified excerpts
  map_store: secure mapping (entity -> pseudonym) stored separately

1) Detect PII spans:
   spans := []
   for x in X:
     spans_x := UnionDetect(x.text, pii_detectors)  # merge overlapping spans
     spans.append((x.excerpt_id, spans_x))

2) Canonicalize + cluster entities for consistency:
   ents := ExtractEntityStrings(spans)
   clusters := ClusterByStringSimilarity(ents, method=rapidfuzz, threshold=t)

3) Assign replacements:
   for cluster in clusters:
     pseud := MakePlaceholder(cluster.type)  # e.g., [PERSON_3], [ORG_2]
     StoreSecure(map_store, cluster -> pseud)  # access-controlled

4) Apply replacements:
   for x in X:
     x'.text := ReplaceSpans(x.text, spans_x, map_store, policy)
     x'.doc_id := HashOrRemap(doc_id)  # optional
   return X'
```

### I.4. A simple code-relation induction algorithm: traceable pairwise induction + graph sparsification

**Goal.** To move beyond treating codes as independent labels (**D1**) and instead induce explicit inter-code relations (axial coding), we need algorithms that operationalize the transition from coding datapoints to modeling relations. Below is a lightweight baseline that (i) proposes candidate relations via simple statistics/ordering cues, (ii) types and orients them using an LLM judge *with evidence*, and (iii) produces a sparse, auditable graph ready for **D2**/**D3**/**D4** representations.

**Inputs.** Assume a coded corpus where each excerpt $x$ has a set of codes $C(x) \subseteq \mathcal{C}$ (from a codebook), and (optionally) a time index.

**Step A: candidate generation (fast, deterministic).** Compute for each code pair $(a, b)$:

- **Co-occurrence strength:** PMI or log-odds based on counts over excerpts:

$$\text{PMI}(a, b) = \log \frac{p(a, b)}{p(a)p(b)}.$$

- **Temporal precedence (if time exists):** within each document, count how often $a$ appears before $b$ within a window $w$:

$$\text{Prec}(a \to b) = \frac{\#(a \text{ before } b)}{\#(a \text{ before } b) + \#(b \text{ before } a) + \epsilon}.$$

Generate candidate edges by thresholds (e.g., PMI $> \tau_{\text{pmi}}$ for association; precedence $> \tau_{\text{prec}}$ for NEXT-like direction).

**Step B: relation typing and direction via evidence-anchored LLM judgments.** For each candidate pair, retrieve a small evidence set:

- **Support candidates:** excerpts where both codes occur (or occur in close temporal proximity).

- **Counterevidence candidates:** excerpts where $a$ occurs without $b$, or where order reverses.

Ask the judge to classify relation type from a small ontology:

$$\{\text{NEXT, CO\_OCCURS, ENABLES, INHIBITS,}$$
$$\text{CAUSES, PART\_OF, MODERATES, NONE}\}.$$

Store its decisions as EvidenceItems on the induced edge.

**Step C: graph sparsification (keep it interpretable).** To avoid "hairball" graphs:

- Keep top-$K$ edges per node by support score.

- For **D2** candidates, apply transitive reduction on NEXT edges (when acyclic) to keep only necessary order constraints.

- For **D4** candidates, detect cycles and aggregate them into Loop objects; compute loop sign if edges are signed.

**What this buys us immediately.** This baseline directly instantiates the paper's call for "code relation induction algorithms" that explicitly move from open coding to inter-code relations (axial coding), and it produces artifacts that are (i) graph-native (usable for **D2/D3/D4**) and (ii) evidence-linked (scorable by the goodness-of-fit pipeline above), while acknowledging that sensitive corpora require de-identification and careful governance.

