# OpenReview forum: "Position: Bridge Human Interpretation and Machine Representation With Explicit Specification For Qualitative Data Analysis In LLM Era"
_ICML.cc/2026/Position_Paper_Track — ICML 2026 Position Paper Track regular_

### Official Review · Reviewer_edFr · 2026-02-23

**Significance:** 3
**Argument Clarity:** 4
**Rating:** 5
**Confidence:** 3

**Questions:**

What credible alternatives do the authors consider besides their 2-axis framework?

Can the authors contextualize some of the differences in their 4x4 grid within an example, e.g. showing failure modes of LLM pipelines for the same rough data across the grid?

**Alternative Views Section:**

Yes

**Compliance With Llm Reviewing Policy A Conservative:**

Affirmed.

**Discussion Potential:**

3

**Final Justification:**

I thank the authors for their rebuttal and maintain my positive score.

**Paper Summary:**

The authors introduce two axes (meaning making & modeling) for specifying what kinds of process LLM-based pipelines intend to produce in qualitative data analysis. They uses these two axes to make a 4x4 grid of levels and use it to compare qualitative outputs and conduct a systematic review across both human-led studies and LLM-assisted workflows. They find that current LLM pipelines fail to leverage parts of this grid that are useful. They then propose a research agenda to help remedy these shortcomings in LLM-assisted qualitative data analysis.

**Position:**

Yes

**Position In Title:**

Yes

**Related Work:**

4

**Strengths And Weaknesses:**

- Strengths
  - The paper tackles an important problem, as LLMs become increasingly widespread in qualitative data analysis
  - The paper is clearly written
  - The paper very thoroughly addresses related work and contextualizes it within the proposed framework
  - The proposed framework is clear and intuitive
- Weaknesses
  - The two-axis framework is slightly abstract, and could benefit from a worked through example to show how it can be made useful to understand and compare analyses
  - Additionally, the authors could consider alternative minimal axes rather than meaning-making and modeling in their alternative views section

**Support:**

4

---

> ### Author Rebuttal · Authors · 2026-03-31
>
> We thank the reviewer for the thoughtful feedback and are glad the reviewer found the problem important, the paper clearly written, and the proposed framework intuitive. We solve the concerns as below
> ## A concrete worked example.
> We have already provided examples in the appendix to explain why a certain paper can be categorized into a certain meaning-making or modeling level.
> To further supplement the concrete worked example, we further provide another instance of conducting our framework on analyzing the failed multi-agent system’s failure traces:
> - D1 would produce a static taxonomy such as “planning drift,” “verification omission,” and “tool misuse.”
> - D2 would organize the same material into stages to describe MAS workflow, for example: task decomposition → retrieval → synthesis → verification, showing where failures tend to arise.
> - D3 would go further and propose a mechanism, such as: ambiguous decomposition leads to inconsistent subgoals, which leads to weak retrieval checks, which in turn produces unsupported synthesis.
> - In D4 (though current MAS rarely reach this level), we may say that task framing, retrieval quality, verification pressure, confidence, and coordination load are jointly evolving state variables: each is updated over time, each has multiple inbound and outbound influences, and the failure pattern is explained by the system’s overall feedback structure rather than by a single directed chain.
>
> This is the kind of distinction the paper is trying to make when it separates static maps, stages, causal pathways, and system dynamics.
> We can also hold the modeling level fixed and vary the meaning-making level.
> - At a causal level, M1 would remain descriptive (“retrieval was skipped before the final answer was produced”),
> - M2 would categorize recurring causal patterns across traces,
> - M3 would introduce an interpretive claim (“the system implicitly prioritized answer completion over verification”), and
> - M4 would reframe the same mechanism through an external lens such as coordination failure, distributed cognition, or organizational sensemaking.
>
> This kind of example demonstrates our claim that the same rough phenomenon can be expressed as different model forms and with different strengths of semantic commitment.
> In revision, we will add this worked example into the paper. The paper already uses this family of settings as a motivating case and also cites failure typologies, phase narratives, and feedback explanations as qualitatively distinct output forms.
> ## Credible alternatives to the two-axis framework.
> Our intent is not to claim that the proposed landscape is the only valid organizing scheme. Rather, we see it as a specification layer that sits alongside several existing alternatives already acknowledged in the paper.
>
> First, there are **method-centered taxonomies** from qualitative research, such as thematic analysis, grounded theory, IPA, discourse analysis, narrative analysis, framework analysis, and ethnography. These are highly credible and already well established, but they primarily name research traditions and procedures rather than explicitly specifying what semantic commitment an output makes and what representational form it takes. That is precisely why we argue for classifying “what the analysis commits to” rather than “what the method is called.”
>
> Also, there are **tradition-specific representational alternatives**, including process tracing, mechanism-based explanation, configurational approaches such as QCA, stage/process narratives, and system-dynamics style accounts. We view these not as rejected competitors, but as important anchors that each illuminate part of the design space. Our framework tries to generalize across them: process tracing and mechanism-based explanation help motivate D3-style causal commitments, while system dynamics motivates D4-style feedback commitments. We will revise the paper to make this comparison explicit and to clarify that the 2-axis framework is intended as a cross-cutting specification language, not a replacement for those traditions.
>
> We selected the present two-axis formulation because it offers a compact interface that cleanly separates semantic inference from representational structure, supports comparison across qualitative outputs, and is sufficiently explicit to be operationalized in annotation and governance settings. This matches the paper’s broader goal of making qualitative analytic commitments explicit rather than leaving them implicit in bespoke pipelines.

---

> > ### Author Rebuttal · Reviewer_edFr · 2026-04-05
> >
> > I thank the authors for their rebuttal and maintain my positive score.

---

### Official Review · Reviewer_9H3L · 2026-03-01

**Significance:** 2
**Argument Clarity:** 2
**Rating:** 4
**Confidence:** 3

**Questions:**

if D4 is so rare even in human QR, is it a realistic target for automation? The call to action for NLP researchers to aim for D4 may set an aspiration that is out of reach given current LLM capabilities and the inherent difficulty of inducing feedback models from textual traces. A more measured discussion of which levels are currently feasible versus long-term goals would strengthen the agenda.

**Alternative Views Section:**

Yes

**Compliance With Llm Reviewing Policy A Conservative:**

Affirmed.

**Discussion Potential:**

3

**Paper Summary:**

This paper introduces a conceptual framework—a 4×4 landscape—for characterizing qualitative research outputs along two orthogonal dimensions: (1) level of meaning-making (descriptive, categorical, interpretive, theoretical) and (2) level of modeling (static taxonomy, stage/timeline, causal pathway, dynamical systems).  The paper concludes with a call to action for both NLP/LLM researchers and qualitative methodologists.

**Position:**

Yes

**Position In Title:**

Yes

**Related Work:**

2

**Strengths And Weaknesses:**

styrengths:
The paper addresses a genuinely pressing issue: as LLMs are increasingly deployed to automate qualitative analysis, the field lacks a shared vocabulary for specifying what kind of analysis is being automated and what representational commitments the outputs carry.

The paper does not merely propose a taxonomy and stop; it conducts a substantial annotation study (300 papers) to demonstrate that the framework reveals systematic patterns. The paper does not merely propose a taxonomy and stop; it conducts a substantial annotation study (300 papers) to demonstrate that the framework reveals systematic patterns. \

Section 5 translates the observed gaps into concrete research directions for both communities. For NLP researchers, the call to move beyond semantic aggregation to relation induction (axial coding) and to treat modeling levels as design targets rather than incidental outputs is specific and technically meaningful. For qualitative methodologists, the call to produce shareable gold artifacts (codebooks, causal maps) and to establish governance rules for when automation is appropriate addresses real concerns about trust and rigor.

 Appendix provides unusually detailed protocols for implementing the framework: a typed graph standard (QualGraph), evidence-anchored fitness metrics, de-identification strategies, and a simple code-relation induction algorithm.

Weakness:

While the authors report 85% agreement for meaning-making and 94% for modeling in the operationalizability experiment, these numbers come from a small sample (35 papers) and use a simplified setup where the LLM was provided only with level definitions.

The empirical QR corpus, while large in initial screening (664k records), ultimately includes only 231 manually annotated papers. The authors acknowledge that the distributions "might not be perfectly unbiased population estimates," but the sampling strategy (first 3 pages of Google Scholar results per query) likely introduces visibility bias toward highly cited or recently published work.

The definition of Level 4 modeling (dynamical systems/feedback) is appropriately stringent, but the paper's own examples of D4 are sparse, and the annotation guidelines acknowledge that true D4 is rare.

The paper repeatedly emphasizes that different cells imply "different validity demands and evaluation targets," but it does not systematically elaborate what those evaluation criteria should be. For example, how should we evaluate a D3 causal graph induced by an LLM?

**Support:**

3

---

> ### Author Rebuttal · Authors · 2026-03-31
>
> We sincerely appreciate the reviewer’s detailed reading and encouraging assessment of the paper’s significance, particularly the recognition that the work addresses a pressing need for a shared vocabulary around automated qualitative analysis. We are also thankful that the reviewer highlighted the substantial annotation study, the concrete research agenda in Section 5, and the unusually detailed implementation protocols in the appendix, and we clarify below the points regarding sampling, evaluation, and the role of D4.
> ## Limited scope of the 35-paper operationalizability experiment.
>
>
> We randomly sampled and annotated 35 papers out of 300 papers in the corpus, which represent (11.67%) of the data size. This is a **common and  reasonable practice in human annotation for validating quality and computing Inter-Rator Reliability** [1,2]. Besides, our purpose is not to show that an LLM can robustly perform qualitative analysis at scale. The goal is to test whether the proposed rubric is sufficiently explicit, coherent, and decision-consistent to be applied by an independent agent. We will revise Sec. 3.4 to clarify this scope and tone down wording that could be read as broader capability validation. We will also surface the Appendix H agreement statistics (including Cohen’s κ) in the main text, so the result is clearly interpreted as evidence of rubric operationalizability rather than model capability.
>
> For prompt, we provide a comprehensive prompt to define the different levels and the boundary between different levels of meaning making and modeling. We keep the prompt same as the rubrics used in human annotation to have consistent human/LLM annotation results.
>
> Overall, we conducted a five-turn iterative process to address annotation disagreement, and finally achieved the cohen kappa score as below.
> - Meaning-Making : 0.744 (Substantial Agreement)
> - Modeling: 0.906 (Almost Perfect Agreement)
>
> [1] Cemri, Mert, et al. "Why do multi-agent llm systems fail?." NeurIPS 2025.
>
> [2] Shen, Hua, et al. "Mind the Value-Action Gap: Do LLMs Act in Alignment with Their Values?." EMNLP  2025.
>
> ## Sampling bias in the empirical QR corpus.
> We acknowledge that the empirical corpus is shaped by query design and by visibility in Google Scholar, and therefore should not be interpreted as an unbiased estimate of the full qualitative-research landscape. Our intent is diagnostic rather than population-estimating: the audit is meant to reveal broad, visible patterns in how qualitative outputs are currently represented, not to claim exact field-wide frequencies. We will revise the paper to make this diagnostic intent more explicit and to foreground the limitations of the search and retrieval strategy more prominently.
> ## Rarity of D4 and realism as an automation target.
> It’s true that D4 is rare in our audit partly because we define it strictly: it requires explicit state-update, feedback, and iteration semantics, rather than rhetorical references to “cycles” or “loops.” D4 is included as an upper-bound representational target in the landscape, not as a claim that it is already common in human qualitative research or readily achievable by current LLM systems. Scarcity in our sampled paper corpus should not be read too literally as scarcity in all qualitative scholarship: in traditions such as history and anthropology, D4-style reasoning may be more common in book-length or long-duration work, where complex feedback processes are difficult to fully represent in a single article. More broadly, our intent is not to suggest that current systems should uniformly target D4. Rather, we expect feasibility and reliability to vary by commitment level, with lower-level operations more tractable in the near term and higher-level semantic/representational commitments progressively more fragile. We will revise the agenda accordingly to distinguish near-term feasible targets from longer-term aspirational ones.
>
> ## Evaluation criteria were not explicit enough.
> The **Appendix I** has already contained a more detailed evaluation sketch, but we appreciate the point that the evaluation story should be more concrete in the main text. Our intended position is that evaluation is cell-dependent: different combinations of meaning-making and modeling imply different validity demands, and outputs should therefore not be judged by a single generic notion of plausibility. We will make this more explicit through concrete examples. For instance, D1 outputs should be judged primarily by structural coherence and coverage; D2 outputs by temporal segmentation and ordering fidelity; D3 outputs by evidence-linked mechanism support, directional warrant, and contradiction rate; and D4 outputs by whether feedback and iterative-update claims are genuinely supported by the corpus. In particular, a D3 causal graph should be evaluated in terms of evidence-linked support for proposed mechanisms and directions of influence, rather than only surface plausibility.

---

> > ### Author Rebuttal · Reviewer_9H3L · 2026-04-06
> >
> > I have read the authors' rebuttal and thank them for the comprehensive responses.

---

### Official Review · Reviewer_xNA1 · 2026-03-11

**Significance:** 2
**Argument Clarity:** 2
**Rating:** 4
**Confidence:** 3

**Questions:**

Please see weaknesses.

**Alternative Views Section:**

Yes

**Compliance With Llm Reviewing Policy A Conservative:**

Affirmed.

**Discussion Potential:**

3

**Final Justification:**

The rebuttal overall address my concerns. I decide to maintain my score.

**Paper Summary:**

This position paper's core contribution is a 4×4 conceptual landscape that disentangles two orthogonal dimensions of qualitative research. Through a structured audit of 300 papers, the authors show that current LLM pipelines are mostly confined to shallow modeling levels (D1-D2) despite richer semantic claims, while human-led qualitative studies often make deep interpretive commitments but lack formalized representational rigor.

**Position:**

Yes

**Position In Title:**

Yes

**Related Work:**

3

**Strengths And Weaknesses:**

Strengths:
1. The paper connects the quantitive research with NLP with its proposed theoretical framework.
2. The paper's position is empirically grounded.
3. The paper gives actionable suggestions for both NLPer and quantitive researcher.

Weaknesses:
1. The empirical audit might be bias due to sampling. For example, Google Scholar might be underrepresent non-English qualitative research.

**Support:**

3

---

> ### Author Rebuttal · Authors · 2026-03-31
>
> We are grateful to the reviewer for **highlighting the core contribution of our 4×4 conceptual landscape** and for recognizing that **the paper connects qualitative research and NLP in an empirically grounded way**. We especially appreciate the **positive assessment that the paper offers actionable guidance for both NLP researchers and qualitative researchers**, and we respond below to the concern.
> ## Potential sampling bias in the empirical audit
> We also appreciate the concern regarding potential sampling bias in non-English work. Our intention is not to present the audit as an unbiased census of all qualitative research globally. Rather, the **audit is meant to play a diagnostic role**: it provides structured empirical evidence about visible current practice and helps reveal the deployment gap that motivates the paper’s position. In that sense, the empirical annotation is supplementary support for the paper’s main contribution (the conceptual framework itself), rather than the primary contribution of the paper.
>
> At the same time, our paper selection followed a rigorous process combining the PRISMA method [1,2] and snowball sampling process [3,4], which are two representative and widely used methods in paper collection and position papers [5,6,7].
> - For the **empirical corpus (231 papers)**: we used a PRISMA-style staged screening pipeline and constructed a broad search space by pairing 4,493 domain keywords spanning 51 domains with 10 qualitative methodology terms, yielding 44,930 query combinations. We then retrieved results at scale from Google Scholar, applied staged screening and eligibility filtering, and drew a simple random sample from the eligible pool for manual annotation.
> - For the **computational corpus (69 papers)**: because many relevant papers do not self-identify cleanly through keywords alone, we complemented this with a snowball-sampling process based on seed queries and backward/forward citation tracing. We will revise the paper to make these details more visible in the main text, including clearer reporting of the coverage over years, keywords, domains, and selection stages, so that readers can better assess the breadth and fairness of the collection procedure.
>
> That said, we fully agree that the corpus is not exhaustive and that query-dependent retrieval, platform visibility, and language coverage can still shape what enters the sample. We will therefore revise the wording to avoid any implication of population-level representativeness or full lack of bias. We will also add a clearer limitation statement that expanding to multilingual sources and broader databases is an important direction for future work.
>
> Reference:
>
> [1] Page, Matthew J., et al. "The PRISMA 2020 statement: an updated guideline for reporting systematic reviews." bmj 372 (2021).
>
> [2] Stefanidi, Evropi, et al. "Literature reviews in HCI: A review of reviews." CHI. 2023.
>
> [3] Goodman, Leo A. "Snowball sampling." The annals of mathematical statistics (1961): 148-170.
>
> [4] Parker, Charlie, Sam Scott, and Alistair Geddes. "Snowball sampling." SAGE research methods foundations (2019).
>
> [5] Shen, H., Knearem, T., Ghosh, R., Alkiek, K., Krishna, K., Liu, Y., ... & Jurgens, D. (2025). Position: Towards Bidirectional Human-AI Alignment. In The Thirty-Ninth Annual Conference on Neural Information Processing Systems Position Paper Track.
>
> [6] Lee, M., Gero, K. I., Chung, J. J. Y., Shum, S. B., Raheja, V., Shen, H., ... & Siangliulue, P. (2024, May). A design space for intelligent and interactive writing assistants. In Proceedings of the 2024 CHI Conference on Human Factors in Computing Systems (pp. 1-35).
>
> [7] Wei, K., Paskov, P., Dev, S., Byun, M. J., Reuel, A., Roberts-Gaal, X., ... & Deshpande, C. (2025). Position: Human baselines in model evaluations need rigor and transparency (with recommendations & reporting checklist). In Forty-second International Conference on Machine Learning Position Paper Track.

---

> > ### Author Rebuttal · Reviewer_xNA1 · 2026-04-02
> >
> > My concerns have been addressed and I decide to maintain my current score.

---

### Official Review · Reviewer_7qNL · 2026-03-11

**Significance:** 2
**Argument Clarity:** 1
**Rating:** 2
**Confidence:** 4

**Questions:**

My questions are implicit in the weaknesses I described above.

**Alternative Views Section:**

Yes

**Compliance With Llm Reviewing Policy A Conservative:**

Affirmed.

**Discussion Potential:**

3

**Final Justification:**

I am electing to stick with my initial assessment. I have real concerns about the core claims of this paper and their connection with ICML. I also feel persistently unclear about the precise nature of the experiments and what what the results would tell us. I recognize that the other reviewers are more positive, but I don't necessarily feel positioned to try to change their minds, so my own less positive position can be balanced against theirs.

**Paper Summary:**

This position paper is broadly about the role, or potential role, for qualitative research (QR) methods in NLP and AI.  I believe that, ultimately, the paper is seeking to offer advice from QR about how LLM-based QR methods could productively be used to improve AI systems. The paper develops a framework of definitions for thinking about the goals/outputs of these potential QR systems, and it uses this framework framework to annotate and analyze a sample of QR papers, assessing them to understand the extent to which they cover all the definitions in the framework. the overall finding of this process is that most of the papers in the research sample are fairly shallow relative to what we might hope for from QR analyses.

**Position:**

Yes

**Position In Title:**

Yes

**Related Work:**

4

**Strengths And Weaknesses:**

Strengths

1. I appreciate the call for more QR using LLMs. I think they can often be more creative and consistent than humans, and they can of course do much more work.

2. The paper seems to be taking its own advice: the annotation project in the paper is, of course, itself an LLM-based QR analysis that depends heavily on expert created prompts and analyses of the outputs. puts, this seems healthy to me and it looks like the evidence is quite valuable.

Weaknesses

1. I found the paper very difficult to follow. It took me a while even to figure out precisely what its audience is and where it is offering its advice. My current understanding is that the paper is entirely about using LLMs to do QR are in diverse domains. It is not about using QR to understand AI systems themselves. The paper is also highly repetitive, and this too makes it difficult to follow because very often old information from previous sections seems to be presented as though it were new. For example, compare section 2.3 with the start of section 3.

2. The above orientation makes me question whether ICML is the right outlet for the work. work. It seems like this research should be oriented toward people who are likely to use LLMs to analyze data from other domains. domains. This is unlikely to be people who are attending ICML. A paper on using QR to understand AI systems would, by contrast, be appropriate for ICML.

3. Although there is an extensive amount of information in the appendices, I was unable to figure out precisely which set of papers was used for the annotation project. This is important to me, again, for understanding the actual focus of the paper. I suspect that if I saw the list of papers, or even the keywords (which are promised in the text but I cannot find in the appendices), this would further compound my concern that this paper is oriented toward a different audience from the ICML one.

**Support:**

2

---

> ### Author Rebuttal · Authors · 2026-03-31
>
> Thank you for the thoughtful and constructive feedback. We are especially encouraged that **you find our core idea—LLM-supported qualitative research (QR)—valuable and that the annotation study is informative**. We agree that **clarity of positioning and presentation can be significantly improved**. We address all the concerns as below and are incorporating all clarifications into the reviewed version.
> ## 1. ICML audience / positioning
> We appreciate this important concern and would like to clarify that the paper is not primarily about applying LLMs to qualitative research. Rather, our central claim is: the paper is about **how LLM-based reasoning pipelines should be structured, represented, evaluated, and interpreted, using qualitative research (QR) as a methodological lens.**
> In modern ML, QR-like analysis is already implicitly present in many core areas:
> - RLHF and preference modeling and optimization
> - Agent trace analysis
> - Human-in-the-loop evaluation
> - Failure analysis of multi-agent systems
>
> However, these analyses are often: (1) underspecified; (2) methodological inconsistent; (3) difficult to evaluate or reproduce due to large amounts of data. Our contribution is to address these issues by **making this QR data analysis space explicit and structured via the meaning-making x modeling framework**. We further clarify this paper’s concrete contributions and relevance to the ICML conference below.
>
> ## 2. Concrete ICML contribution and relevance
> To clarify relevance to ICML, LLM-based QR directly supports several core ML problems:
> ### 2.1 Data, labeling, and bias
> Prior work shows bias emerges from human–system interaction [1,2]. LLM-based QR complements this by:
> - structuring annotator discussions and disagreements
> - identifying latent failure modes in data pipelines
> - moving beyond distributional metrics to process-level explanations
>
> [1] Data Feedback Loops: Model-driven Amplification of Dataset Biases. ICML 2023
>
> [2] Position: When Incentives Backfire, Data Stops Being Human. ICML 2025
>
> ### 2.2 Human feedback and alignment
>  While RLHF/DPO/PPO focus on preference labels [3,4]. LLM-based QR analyzes:
> free-text rationales
> - disagreement patterns
> - implicit annotator norms
>
> This provides richer signals for process-based reward modeling, enabling alignment on open-ended tasks (e.g., via rubric-based evaluation) and improving fidelity to human values.
>
> [3] Language Instructed Reinforcement Learning for Human-AI Coordination. ICML 2023
>
> [4] Comparing Comparisons: Informative and Easy Human Feedback with Distinguishability Queries, ICML 2025
>
> ### 2.3 Human-in-the-loop evaluation
> Recent work highlights limits of outcome-only metrics  [5,6]. LLM-based QR enables:
> - modeling evaluator reasoning
> - recovering coordination failures
> - improving process-level evaluation signals
>
> [5] A Mathematical Framework for AI-Human Integration in Work, ICML, 2025
>
> [6] Agent-as-a-Judge: Evaluate Agents with Agents, ICML 2025
> ### 2.4 Multi-agent system analysis
> Qualitative taxonomies are key to understanding failures [7,8]. LLM-based QR generalizes this by:
> - turning trajectories into causal accounts of failure
> - identifying when, why, and how coordination breaks down
>
> [7] Why Do Multi-Agent LLM Systems Fail? NeurIPS 2025
>
> [8] AgenTracer: Who Is Inducing Failure in the LLM Agentic Systems? ICLR 2026
> ### Summary
> Our framework is domain-agnostic and is directly relevant to understanding AI systems. While the current paper emphasizes qualitative research (QR) in external application domains to illustrate the broader space of qualitative outputs, the same framework also applies to core ML questions such as analyzing LLM reasoning, interpreting agent trajectories, and studying emerging model behaviors. More broadly, it can help structure human understanding, behavior, and values in ways that are useful for alignment and safety, by connecting human feedback and preferences to model representations, as illustrated in Figure 1. We will revise the paper to foreground this ML-centric framing more explicitly.
>
> ## 3. On annotation corpus transparency
> We thank the reviewer for highlighting this. The annotation study is based on 231 empirical QR papers and 69 computational papers. We will revise to:
> - state this clearly in the main paper (not only appendix)
> - explicitly list selection criteria and keywords
> - improve pointers to the appendix and repository
> - increase visibility of the annotated corpus
>
> We also provide an anonymized repository of annotated papers:
>
> https://anonymous.4open.science/r/QA-Papers-64F5/XXXX-3.md
>
> The computational corpus spans keywords such as:
> - Structured information extraction foundations
> - Event schemas, tracking, and prediction
> - Interactive human-AI qualitative analysis tools
> - Deductive coding and codebook-guided labeling
> - Inductive thematic analysis and grounded theory
> - Multi-agent automation for thematic analysis
> - Topic modeling, concept induction, and RAG-based sensemaking

---

> > ### Author Rebuttal · Reviewer_7qNL · 2026-04-02
> >
> > I appreciate the authors' reply, which does clarify the goals of the paper. However, I still found the paper difficult to follow, which leads me to want to see a substantial rewrite before I recommend acceptance. In addition, while I don't want to be a gate-keeper here, this still does not seem well aligned with ICML, since this seems to be a paper about practical applications of LLMs to other research topics.

---

### Decision · Program_Chairs · 2026-04-30

**Decision:**

Accept (regular)

**Comment:**

The paper presents a compelling argument for rethinking the approach to qualitative data analysis with LLMs. The paper provides convincing evidence and empirical results, as well as clear calls to action for multiple audiences. The paper has some organization/presentation issues and the readability could be improved by adding concrete examples to help understand its abstractions.